# Acetylation of H3K115 is associated with fragile nucleosomes at CpG island promoters and active regulatory sites

Yatendra Kumar[1][*][†], Dipta Sengupta[1][†], Elias T Friman[1], Robert S Illingworth[2], Manon Soleil[3], Zheng Fan[4], Hua Wang[5], Kristian Helin[4], Matthieu Gérard[3], Wendy A Bickmore[1][*]

[1]MRC Human Genetics Unit, Institute of Genetics and Cancer, University of Edinburgh, Edinburgh, United Kingdom; [2]MRC Centre for Regenerative Medicine, Institute for Regeneration and Repair, University of Edinburgh, Edinburgh, United Kingdom; [3]Université Paris-Saclay, CEA, CNRS, Institute for Integrative Biology of the Cell (I2BC), Gif-sur-Yvette, France; [4]Division of Cancer Biology, The Institute of Cancer Research, London, United Kingdom; [5]Department of Lymphoma, Peking University Cancer Hospital and Institute, Peking University International Cancer Institute, Peking University Health Science Center, Beijing, China

**\*For correspondence:**
yaten2020@gmail.com (YK);
wendy.bickmore@ed.ac.uk
(WAB)

[†]These authors contributed
equally to this work

**Competing interest:** The authors
declare that no competing
interests exist.

**Reviewing Editor:** Jerry L
Workman, Stowers Institute for
Medical Research, Kansas City,
United States

## eLife Assessment

This **fundamental** study provides the first genome-wide characterization of H3K115 acetylation and identifies a striking and previously unappreciated association of this globular-domain histone modification with fragile nucleosomes at CpG island promoters, active enhancers, and CTCF binding sites. While the work is largely descriptive and correlative in nature the evidence is **compelling**. The authors present multiple, orthogonal genomic and biochemical analyses that consistently support their central conclusions.

**Abstract** Acetylation of lysine residues in the tail domain of histone H3 is well characterised, but lysine residues in the histone globular domain are also acetylated. Histone modifications in the globular domain have regulatory potential because of their impact on nucleosome stability but remain poorly characterised. In this study, we report the genome-wide distribution of acetylated H3 lysine 115 (H3K115ac), a residue on the lateral surface at the nucleosome dyad, using chromatin immunoprecipitation. In mouse embryonic stem cells, we find that detectable H3K115ac is enriched at the transcription start site of active CpG island promoters, but also at polycomb-repressed promoters prior to their subsequent activation during differentiation. By contrast, at enhancers, H3K115ac enrichment is dynamic, changing in line with gene activation and chromatin accessibility during differentiation. Most strikingly, we show that H3K115ac is detected as enriched on 'fragile' nucleosomes within nucleosome-depleted regions at promoters and active enhancers, where it coincides with transcription factor binding, and at CTCF-bound sites. These unique features suggest that H3K115ac correlates with, and could contribute to, nucleosome destabilisation and that it might be a valuable marker for identifying functionally important regulatory elements in mammalian genomes.

## Introduction

Post-translational modifications (PTMs) of histone tails play key roles in chromatin structure and gene regulation. Modifications of the histone tails have attracted attention due to the unstructured nature of the domain (*Ghoneim et al., 2021*; *Peng et al., 2021*) and their accessibility to chromatin modifiers and readers. However, more recently, modifications in the globular domains of core histones have been associated with various chromatin functions (*Tropberger and Schneider, 2013*; *Pradeepa et al., 2016*; *Zorro Shahidian et al., 2021*) but remain relatively understudied.

Modifications on the lateral surface of the histone octamer are of particular interest because these can directly affect interactions with DNA (*Cosgrove et al., 2004*; *Lawrence et al., 2016*). For example, acetylation of lysine-56 on histone H3 (H3K56ac) is predicted to increase accessibility at both the entry and exit sites of the nucleosome, impacting gene expression, DNA replication and repair (*Rajagopalan et al., 2017*; *Rodriguez et al., 2019*). Lysine-64 on H3 is close to the nucleosome dyad axis and its acetylation (H3K64ac) decreases nucleosome stability (*di Cerbo et al., 2014*). Lysine-122 of H3 (H3K122) is also located close to the dyad axis and its acetylation impacts nucleosome stability and weakens histone-DNA interactions (*Manohar et al., 2009*; *Rajagopalan et al., 2017*) with a direct consequence on transcription from a chromatinised template in vitro (*Tropberger et al., 2013*). H3K122ac also makes nucleosomes more susceptible to disassembly by chromatin remodellers (*Chatterjee et al., 2015*). Other acylation events, such as succinylation of H3K122, also impact nucleosome stability (*Zorro Shahidian et al., 2021*). Whilst H3.3 is not essential in mouse embryonic stem cells (mESCs), mutation of H3.3 K122 to alanine is cell lethal (*Patty et al., 2024*).

H3K115 is in the globular domain of H3 at the nucleosome dyad. Because both H3K115 copies in the histone octamer are juxtaposed on the lateral surface of the nucleosome in close contact with the overlying DNA (*Figure 1A*), acetylation of H3K115 has a high potential to disrupt histone-DNA interactions and influence nucleosome dynamics, nucleosome breathing, and histone release (*Zhou et al., 2019*). Consistent with this, acetylation of H3K115 (H3K115ac) favours nucleosome disassembly and nucleosome sliding in vitro, by itself or together with H3K122ac (*Manohar et al., 2009*; *Simon et al., 2011*).

Promoters have a specific nucleosome organisation: two well-positioned nucleosomes (+1 and –1) flank a nucleosome-depleted region (NDR) immediately upstream of the transcription start site (TSS). This view has been revised with the detection of micrococcal nuclease- (MNase) and salt-sensitive nucleosome species within the NDRs after limited chromatin digestion with MNase. These have been termed fragile, unstable, or partially unwrapped nucleosomes (*Henikoff et al., 2011*; *Kent et al., 2011*). Fragile nucleosomes can be isolated by MNase digestion under mild conditions and contain the H3.3 and H2AZ histone variants (*Jin and Felsenfeld, 2007*). In mESCs, chemical cleavage methods have also identified subnucleosomes at TSSs (*Ishii et al., 2015*; *Voong et al., 2016*). To the best of our knowledge, fragile nucleosomes have not yet been associated with a specific histone modification in mammalian cells.

H3K64ac and H3K122ac have been detected by chromatin immunoprecipitation (ChIP) at the TSS of active genes and at some active enhancers in mammalian cells (*Tropberger et al., 2013*; *di Cerbo et al., 2014*; *Pradeepa et al., 2016*). In this study, we report the distribution of H3K115ac, as detected by ChIP, across the mammalian genome in mESCs, and during the differentiation to neural progenitor cells (NPCs). This reveals the focal association of H3K115ac with fragile nucleosomes at both active and repressed CpG island (CGI) promoters, at active enhancers, and at other sites important for genome architecture. Our data suggest that H3K115ac ChIP signal has a genomic distribution distinct from other H3 acetylation marks, and that it could be an important new tool for the functional annotation and understanding of mammalian genomes.

## Results

### H3K115ac is associated with active regions with a strong preference for CGI promoters

To determine where H3K115 acetylated nucleosomes can be detected across the mammalian genome, we used a commercially available antibody and confirmed its specificity for H3K115ac and not for unmodified H3K115, H3K115R, or H3K122ac by dot blotting against synthetic peptides

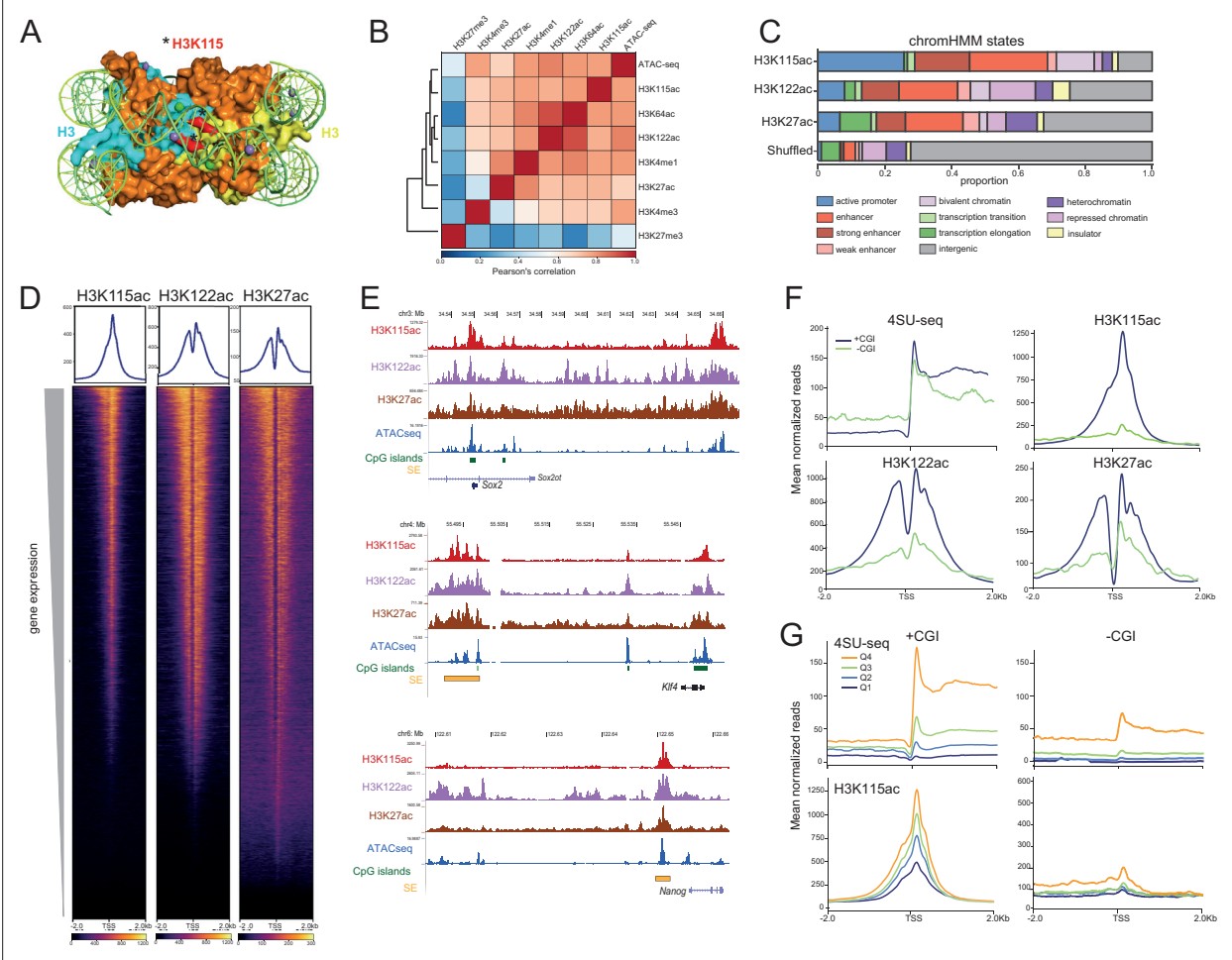

**Figure 1.** H3K115ac is associated with CpG island (CGI) promoters. (**A**) Nucleosome structure looking down on the dyad axis (modified from PDB-5X7X, *Taguchi et al., 2017*). The two H3 molecules are shown in cyan and yellow, other histones are in orange and DNA in green. N-terminal histone tails are hidden. Both copies of H3K115 (red and asterisked) are juxtaposed at the dyad axis close to the overlying DNA. (**B**) Pearson's correlation matrix with hierarchical clustering in mouse embryonic stem cells (mESCs). Correlation is computed for read counts in 10 kb windows across the genome for ATAC-seq data, ChIP-seq data for active (H3K122ac/H3K27ac/H3K27ac: GSE66023; H3K4me3: GSM1003756; H3K4me1: GSM1003750; and repressive H3K27me3: GSM1276707), histone H3 modifications and for H3K115ac. (**C**) Proportions of H3K115ac, H3K122ac, and H3K27ac ChIP peaks that overlap genomic segments defined by chromHMM in the mouse genome (*Ernst and Kellis, 2012*; *Pintacuda et al., 2017*). (**D**) Heatmap of H3K115ac ChIP-seq signal (this study), H3K122ac, and H3K27ac (*Pradeepa et al., 2016*) in mESCs with respect to transcription start site (TSS) of the top 50% of genes by expression and sorted by decreasing gene expression. (**E**) UCSC genome browser screenshot showing ChIP-seq data for H3K115ac, H3K122ac, H3K27ac, and ATAC-seq in mESCs at the Sox2, Klf4, and Nanog loci. CGIs are indicated. Genome co-ordinates (Mb) are from the mm10 assembly of the mouse genome. (**F**) Mean normalised reads of, (upper left) 4SU sequencing (4SU-seq) centred at TSS of top 50% genes by expression (4SUseq tags in TSS+ 500 bp region). Genes are divided into TSS that do (CGI+) or do not (CGI-) overlap with CGIs. Upper right and lower panels show average profiles of H3K115ac, H2K122ac, and H3K27ac ChIP-seq read-density at these same TSS classes. The higher H3K115ac read-density at CGI+ TSS is not due to sample size (Wilcox, p-value<2.2e-16, normalised read coverage within a window spanning TSS+ 500 bp). (**G**) Average profiles (mean normalised reads) of, (top); 4SU-seq in mESCs centred at protein-coding TSSs (±2 kb) divided into quartiles (Q1–Q4) of 4SU-seq signal within 500 bp upstream and downstream of the TSS for (left) TSS overlapping a CGI (+CGI), or (right) promoters without any CGI (-CGI). Below: H3K115ac ChIP-seq signal around these TSS quartiles defined by 4SU.

The online version of this article includes the following figure supplement(s) for figure 1:

**Figure supplement 1.** H3K115ac antibody specificity.

(*Figure 1—figure supplement 1A*) and for a variety of H3 and H4 lysine PTMs by ChIP of bar-coded nucleosomes (*Figure 1—figure supplement 1B*).

Using this antibody, we then performed native ChIP followed by next-generation sequencing (ChIP-seq) from mESCs. H3K115ac ChIP data were highly reproducible between biological replicates (Pearson's correlation r=0.97, *Supplementary file 1*). Genome-wide analysis showed that H3K115ac

ChIP peaks correlate with regions decorated with histone H3 PTMs associated with active gene regulation, both on the H3 tail (H3K27ac, H3K4me1) and at the lateral surface of the nucleosome (H3K64ac and H3K122ac), as well as with regions of chromatin accessibility as assayed by the ATAC-seq data we generated (Pearson's correlation r=0.97 between replicates, *Supplementary file 1*) (*Figure 1B*). Using a chromHMM segmentation approach, we observed that H3K115ac regions have a greater degree of overlap with active promoters and enhancers, compared to H3K122ac and H3K27ac (*Figure 1C*). Notably, fewer peaks are called for H3K115ac (~39,000), compared to other H3 lysine acetylation marks in mESC (H3K122ac~80,000; H3K64ac~65,000; H3K27ac>100,000).

Approximately one third of detected H3K115ac peaks are associated with TSS, and heatmaps show H3K115ac levels generally increasing with the transcription levels as assayed by 4SU sequencing (4SU-seq) (*Benabdallah et al., 2019*; *Figure 1D*). However, while examining key pluripotency genes, we noticed that H3K115ac enrichment, along with that for H3K64ac and H3K122ac, is detected at the TSS of Sox2 and Klf4, but not that of Nanog, despite an ATAC-peak being detected at the Nanog TSS, indicating that H3K115ac detection is not simply a feature of all open chromatin (*Figure 1E*). The Sox2 and Klf4 promoters are located within CGIs, but the promoter of Nanog lacks a CGI, prompting us to investigate H3K115ac with respect to CGIs. Genome wide, ~60% of CGI-associated TSSs (CGI promoters) are detected as associated with H3K115ac in mESCs compared to just 2% for non-CGI promoters. Plotting mean normalised reads around (±2 kb) of TSS shows that CGI promoters exhibit H3K115ac levels many fold higher than non-CGI promoters with comparable levels of transcription (*Figure 1F*). This same discrimination between CGI and non-CGI promoters is not so pronounced for H3K122ac and H3K27ac ChIP data.

Non-CGI promoters have lower overall levels of transcription compared to CGI promoters, and for this promoter class, H3K115ac enrichment detected by ChIP is only really seen for the highest quartile of transcription (4SU) (*Figure 1G*). CGI promoters, on the other hand, exhibit significant levels of detected H3K115ac even for the lowest quartile of expression. These results suggest a special link between CGI promoters and H3K115ac.

## H3K115ac is detected at CGI promoters poised for activation

To determine the relationship between the detection of H3K115ac and dynamic gene expression, we differentiated 46c mESC cells to NPCs monitoring differentiation by Sox1-GFP fluorescence. ChIP-seq for H3K115ac (Pearson's correlation r=0.90, *Supplementary file 1*) and ATAC-seq (Pearson's correlation r=0.94, *Supplementary file 1*) were carried out after 7 days of differentiation and nascent RNA quantified (4SU-seq) on days 0, 3, 5, and 7 (*Benabdallah et al., 2019*).

TSSs were classified into those that gained or lost H3K115ac ChIP signal during differentiation if the TSS (–500 to +500 bp) showed at least a twofold change in H3K115ac signal compared with day 0. Average levels of transcription increased from TSS that also gained H3K115ac, and there was loss of H3K115ac from downregulated TSS (*Figure 2A*).

Given the strong bias of H3K115ac enrichment towards CGI promoters largely independent of transcription (*Figure 1F*), we classified CGI- and non-CGI promoters based on changes in H3K115ac (gain/loss) and expression (up/down) during differentiation (*Figure 2B*). Promoters exhibit positive enrichment (log$_2$observed/expected) for congruent changes of H3K115ac and transcription (gain~up and loss~down) and negative enrichment for incongruent changes (gain~down and loss~up) (*Figure 2B*, *Supplementary file 2a*). However, these enrichments are stronger for non-CGI promoters than for CGI promoters. This is consistent with our observation that some CGI promoters have significant levels of H3K115ac detectable while being transcribed at extremely low levels (*Figure 1G*).

In line with this, we identified a set of promoters that are transcriptionally upregulated in NPC but without any significant changes in the levels of H3K115ac detected (*Figure 2C*). In mESC, these promoters exhibit very low transcription along with high levels of H3K27me3, which are then lost upon differentiation (*Goronzy et al., 2022*). Compared to these promoters, a background set of promoters with no detected changes in H3K115ac and no change in transcription (>2-fold) during differentiation has very low levels of H3K27me3 (*Figure 2—figure supplement 1A*). H3K115ac-marked promoter sets – including those showing no detectable change in H3K115ac despite transcriptional activation during differentiation – are enriched for high-confidence bivalent genes taken from another study (*Seneviratne et al., 2024*, *Supplementary file 2b*). This indicates that H3K115ac, as assayed by ChIP, is a feature of CGI-associated genes that are polycomb targets in mESCs and that are poised for

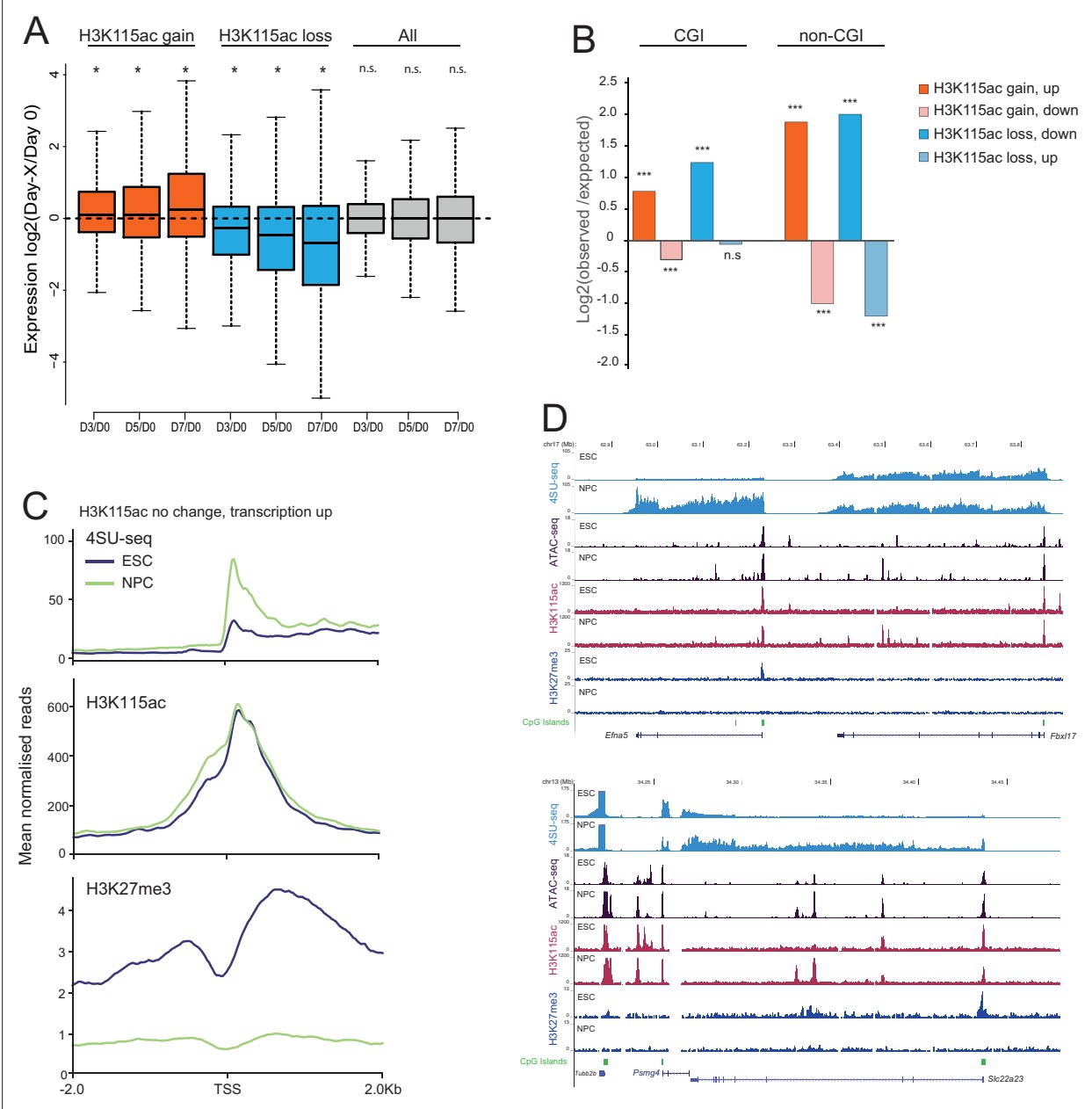

**Figure 2.** Changes in H3K115ac during differentiation. (**A**) Boxplots displaying the changes in activity (4SU sequencing [4SU-seq]) of promoters that gain or lose H3K115ac chromatin immunoprecipitation (ChIP) signal across the 7 days of mouse embryonic stem cell (mESC) to neural progenitor cell (NPC) differentiation. Log$_2$ fold change is shown relative to day 0. *Paired Wilcox, p<0.01. (**B**) Bar plot showing enrichment of gene sets defined based on differential H3K115ac (gain/loss) and differential expression during differentiation (up/down). Enrichments are calculated for CpG island (CGI) and non-CGI promoters. ***Fisher's exact test p<0.01; n.s. p>0.01, *Supplementary file 2a*. (**C**) Aggregate profile plots for 4SU-seq and ChIP-seq data for H3K115ac and H3K27me3 (*Mikkelsen et al., 2007*) from mESCs and NPCs at promoters with no significant change in H3K115ac occupancy, but significant transcriptional upregulation during differentiation. (**D**) UCSC genome browser screenshot showing 4SU-seq, ATAC-seq, and ChIP-seq data for H3K115ac and H3K27me3 (*Mikkelsen et al., 2007*) in mESCs and differentiated NPCs at the Efna5 and Slc22a23 loci. CGIs are indicated. Genome co-ordinates (Mb) are from the mm10 assembly of the mouse genome.

The online version of this article includes the following figure supplement(s) for figure 2:

**Figure supplement 1.** H3K115ac dynamics during differentiation.

later activation in NPCs (*Figure 2—figure supplement 1B*). Examples of this include Efna5, Slca23 (*Figure 2D*), and Insm1 (*Figure 2—figure supplement 1C*), all genes known to have a function in neuronal cells.

H3K115ac shows a stronger enrichment at CGI, polycomb-repressed, and bivalent promoters in mESC than two other globular domain H3 modifications – H3K64ac and H3K122ac (*Figure 2—figure supplement 1D*, *Supplementary file 2c*). H3K27ac, as expected, is depleted from CGI promoters, since H3K27ac and H3K27me3 are mutually exclusive. In contrast to H3K115ac, H3K64ac and H3K122ac are enriched over the gene body and upstream regions, but not at the TSS, of CGI-associated genes (*Figure 2—figure supplement 1E*).

## H3K115ac is associated with fragile nucleosomes within promoter NDRs

We noted that H3K115ac ChIP signal over promoters appears much more focal and centred over the TSS compared with both H3K27ac and H3K122ac that are depleted over the TSS (*Figure 1D*). This suggests that H3K115ac may be associated with MNase-sensitive nucleosomes in the promoter NDR. These nucleosomes can be detected as subnucleosomes (footprint <147 bp) obtained from small fragments sequenced from partial MNase digestion (*Carone et al., 2014*) or mononucleosomes (footprint ~147 bp) obtained under low salt conditions (*Jin and Felsenfeld, 2007*). Both species of nucleosome particles are referred to as fragile nucleosomes (*Carone et al., 2014*; *Ishii et al., 2015*; *Voong et al., 2016*).

To investigate the distribution of H3K115ac in relation to nucleosome size, we repeated the H3K115ac and H3K27ac ChIP on partially MNase-digested native chromatin from mESCs and sequenced both ends of the purified DNA fragments (paired-end) (Pearson's correlation r=0.99 between biological replicates for both H3K115ac and H3K27ac, *Supplementary file 1*). Around (± kb) the most active promoters (4SU Q4), mean signal from small (<150 bp) MNase-digested fragments accumulates inside the TSS NDR (*Figure 3—figure supplement 1A*). Signal from fragments >150 bp exhibits the characteristic nucleosome depletion, with the NDR becoming more pronounced with increasing fragment lengths. We used this rationale to split ChIP fragments into subnucleosomes (≤150 bp) and mononucleosomes (151–230 bp) (*Figure 3—figure supplement 1B*). Paired-end ChIP-seq confirmed H3K115ac detection within the NDR of active promoters on subnucleosomes and mononucleosomes immediately upstream of the TSS (*Figure 3—figure supplement 1C*). In contrast, H3K27ac signal is enriched in the gene body and upstream of the TSS but is depleted over the NDR.

H3K115ac ChIP preferentially selects nucleosomes that are shorter than the average input nucleosomes. This is not the case for nucleosomes selected by H3K27ac ChIP (*Figure 3—figure supplement 1D*, *Supplementary file 3a*). This is an indication that H3K115ac may be associated with destabilised and partially unwrapped fragile nucleosomes. Because nucleosomes associated with A/T-rich sequences are known to be more sensitive to MNase compared to sites of higher G/C content (*Chereji et al., 2019*), we analysed A/T content in H3K115ac and H3K27ac libraries. Sequences associated with H3K115ac sub- and mononucleosomes have significantly higher A/T content than those marked with H3K27ac (*Figure 3—figure supplement 1E*, p<0.01, Wilcox test, *Supplementary file 3b*). This bias does not originate from differences in MNase digestion levels, because the matched input libraries do not show such a difference. This suggests a stronger association of H3K115ac detection with nucleosome fragility compared to H3K27ac. Difference in A/T content also suggests that H3K115ac and H3K27ac fragments come from distinct populations of nucleosomes. H3K115ac-modified subnucleosomes and mononucleosomes have the same A/T content, indicating that H3K115ac nucleosomes are present in different stages of unwrapping or stability (*Figure 3—figure supplement 1E*).

To explore H3K115ac ChIP paired-end fragment length (bp) from mESCs in relation to promoter NDRs and flanking nucleosomes, we plotted fragment length against the distance between the ChIP fragment centre and the nearest TSS as 2D contour V-plots of highest density regions for a conservative TSS set, with colour coding of the plots showing the proportion of ChIP fragments within a particular contour (see Materials and methods). Signal from input libraries shows the promoter NDR flanked by positioned nucleosomes with smaller nucleosome particles of varying sizes inside the NDR (*Figure 3A*). H3K27ac signal is most abundant on the +1 nucleosome with barely detectable signal inside the NDR. In contrast, H3K115ac is most enriched on the –1 nucleosome and on subnucleosome-sized fragments inside the NDR (*Figure 3A*, bottom panel). We then defined NDRs

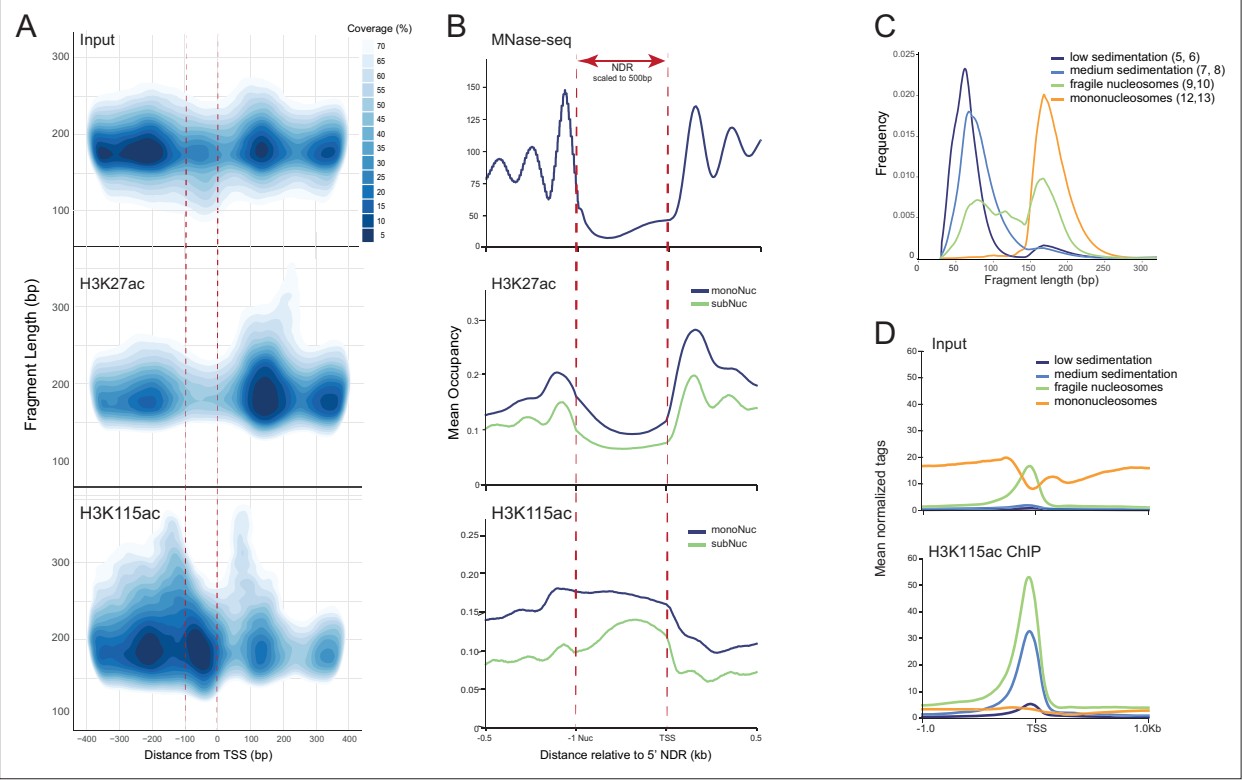

**Figure 3.** H3K115ac is associated with fragile nucleosomes. (**A**) Contour plots depicting the high-density regions of chromatin fragments around transcription start site (TSS) (±400 bp) as a function of fragment length (bp) generated from (top) input MNase library, (centre) H3K27ac chromatin immunoprecipitation (ChIP), and (bottom) H3K115ac ChIP. Coverage refers to the proportion of ChIP fragments in the indicated colour-coded contours. The region from 100 bp upstream to the TSS is indicated with dashed lines in red. (**B**) Mean nucleosome occupancy around mouse TSSs plotted with respect to the nucleosome-depleted region (NDRs), scaled to a length of 500 bp. MNase-seq data (top, *West et al., 2014*) were used to define NDR as the region between the 3' boundary of –1 nucleosome and the TSS. Mean occupancy of (middle) H3K27ac or (bottom) H3K115ac ChIP-seq fragments is split into sub- and mononucleosomes around the scaled NDRs. (**C**) Fragment length distribution (bp) of MNase-digested native chromatin fractionated with sucrose gradient sedimentation. Fractions with different nucleosome species (based on the fragment length) were pooled (indicated in parentheses). (**D**) Input (top) and H3K115ac ChIP-seq (bottom) data, centred at TSS (±1 kb) of most active genes (top 25%) in mouse embryonic stem cells (mESCs), performed on different nucleosome species isolated with sucrose gradient sedimentation from panel **C**. Data is spike-in normalised.

The online version of this article includes the following figure supplement(s) for figure 3:

**Figure supplement 1.** H3K115ac is associated with subnucleosome-sized fragments.

by taking the distance between the TSS and the –1 nucleosome positioning data from published datasets (*West et al., 2014*; *Voong et al., 2016*), scaled them to the same length, and plotted the coverage of stable nucleosomes (MNase-seq), H3K27ac and H3K115ac fragments at these sites (*Figure 3B*). H3K27ac is generally depleted from the NDR, but both H3K115ac-modified mono- and subnucleosomes are enriched within the NDR. Within the NDR, subnucleosome-sized H3K115ac MNase fragments are TSS proximal, whereas mononucleosomal fragments are displaced towards the –1 nucleosome (*Figure 3B*).

To further investigate the properties of subnucleosomal and mononucleosomal particles, we separated MNase-digested mESC chromatin by sucrose gradient sedimentation which can separate fragile and stable nucleosomes (*Nocente et al., 2024*; *Figure 3C*). Sequencing of the chromatin fractions showed the expected depletion of mononucleosomes at the TSS NDR, whereas the slower sedimenting fragile nucleosomes are enriched over the NDR (*Figure 3D*). H3K115ac ChIP-seq from chromatin fractions showed that H3K115ac is detected as enriched in fragile nucleosomal and even slower sedimenting (lower mass) subnucleosomal fractions at TSS, but not in fractions with stable mononucleosomes (*Figure 3D*).

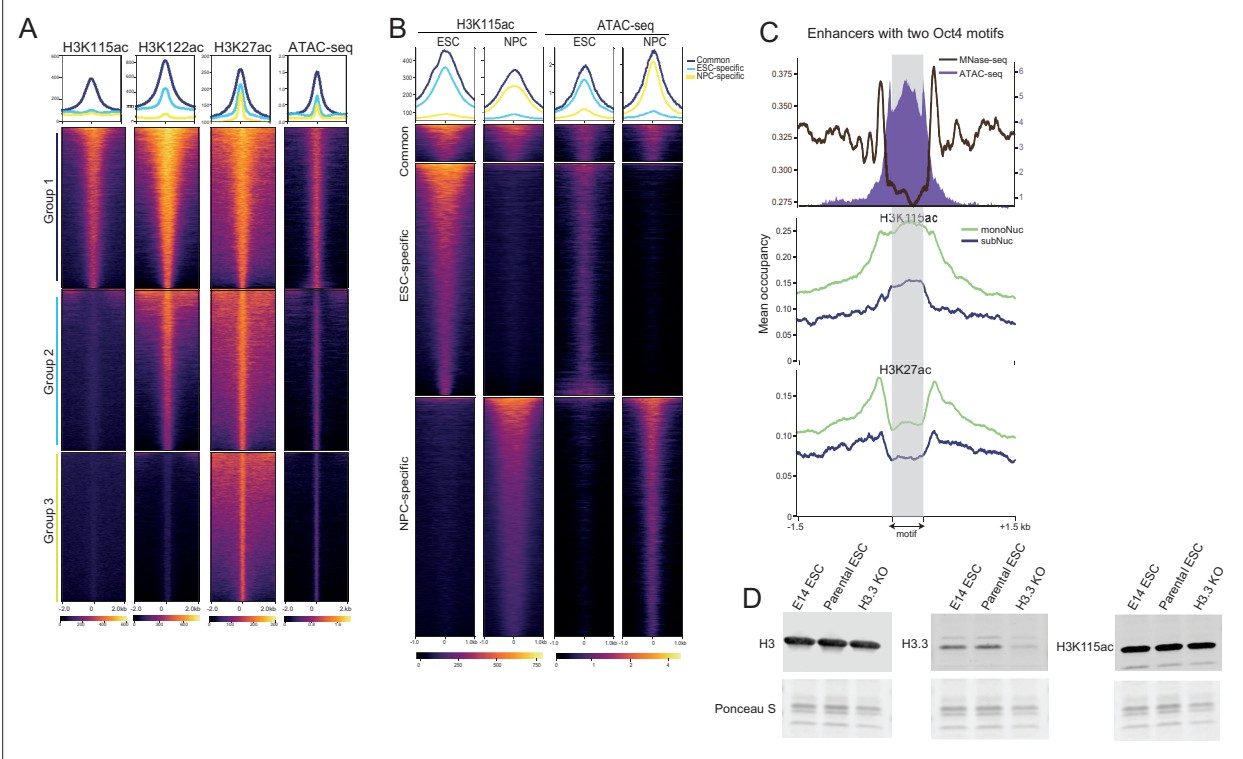

**Figure 4.** H3K115ac marks active enhancers. (**A**) Heatmap showing the coverage of H3K115ac, H3K122ac, H3K27ac, and ATAC-seq centred on promoter distal accessible peaks (putative enhancers) in mouse embryonic stem cells (mESCs). Data is grouped in enhancers marked by all three H3 acetylation marks (Group 1; H3K27ac+ H3K122ac+ H3K115ac+), just H3K27ac together with H3K122ac (Group 2) or H3K27ac alone (Group 3). (**B**) Heatmaps showing H3K115ac and ATAC-seq signal for common and dynamic enhancers between ESC and neural progenitor cell (NPC). Loss/gain of H3K115ac correlates with loss/gain in chromatin accessibility. (**C**) mESC enhancers selected based on the presence of two Oct 4 motifs (n=650) within the Tn5-accessible region, with the region between the two motifs scaled to the same length (shaded grey region). Top: MNase-seq signal (black, left y-axis) and ATAC-seq (purple, right y-axis). H3K115ac ChIP-seq (middle panel) and H3K27ac ChIP-seq (bottom panel) on mononucleosomes (monoNuc; green) or subnucleosome-sized fragments (subMuc; blue). (**D**) Immunoblotting for H3, H3.3, and H3K115ac on whole-cell extracts from E14 mESCs, H3.3 knock-out ESCs, and the parental ESC line. Ponceau S staining of histones is shown below as loading control (*Figure 4—source data 1* and *Figure 4—source data 2*). A biological replicate blot is shown in *Figure 4—figure supplement 1*.

The online version of this article includes the following source data and figure supplement(s) for figure 4:

**Source data 1.** Original immunoblots for *Figure 4D*, indicating the mouse embryonic stem cell (mESC) samples (E14, parental, H3.3KO) and the antibodies used (H4, red; H3, H3.3, and H3K115ac, green).

**Source data 2.** Original files for immunoblots for *Figure 4D*.

**Figure supplement 1—source data 1.** Original immunoblots for lower panel indicating the mouse embryonic stem cell (mESC) samples (E14, parental, H3.3KO) and the antibodies used (H4, red; H3, H3.3, and H3K115ac, green; β-tubulin, green).

**Figure supplement 1—source data 2.** Original files for immunoblots for *Figure 4—figure supplement 1b*.

**Figure supplement 1.** H3K115ac correlates with regulatory activity at mouse embryonic stem cell (mESC) enhancers.

## H3K115ac-modified nucleosomes overlap transcription factor binding sites at active enhancers

With chromHMM segmentation, the largest proportion of H3K115ac ChIP peaks overlap regions identified as putative enhancer elements (*Figure 1C*). Indeed, H3K115ac peaks decorate the known super enhancers of Sox2, Klf4, and Nanog, even though the latter gene lacks H3K115ac at its non-CGI promoter (*Figure 1E*). Peaks detected by ChIP for H3K115ac overlap with a subset of enhancers that are marked by H3K27ac and H3K122ac and that exhibit the highest chromatin accessibility as assayed by ATAC-seq compared to enhancers marked with H3K27ac and H3K122ac, alone and in combination (*Figure 4A*).

Sequences capable of enhancer activity on episomes in mESCs have been identified by STARR-seq (*Peng et al., 2020*), but only half of these correspond to accessible endogenous chromatin sites, as

determined by ATAC-seq. Compared with H3K27ac and various acetyl-lysines on histone H2B (*Narita et al., 2023*), H3K115ac detected by ChIP better discriminates those STARR-seq hits mapping to accessible (ATAC+) sites in the mESC genome from those that are ATAC– (*Figure 4—figure supplement 1A*), even though the proportions of total H3K115ac and H2B acetylated lysine peaks in the mESC genome that are ATAC+ are similar (*Figure 4—figure supplement 1B*). Association of H3K115ac with active enhancers is further substantiated by the enrichment of transcription factors, co-activators, and RNA polymerase II at non-TSS H3K115ac regions as revealed by a random forest approach using published datasets (*Figure 4—figure supplement 1C*).

During mESC to NPC differentiation, the most frequent changes in H3K115ac detection occur at enhancers (*Figure 4—figure supplement 1D*). Loss of H3K115ac signal from ESC-specific enhancers correlates to loss of chromatin accessibility (ATAC-seq), and enhancers gaining H3K115ac signal during differentiation to NPCs also gained chromatin accessibility (*Figure 4B*).

Like promoters, active enhancers are known to be depleted of stable nucleosomes (*Oruba et al., 2020*). We therefore analysed H3K115ac and H3K27ac paired-end ChIP-seq data at mESC enhancers occupied by the pluripotency TF Oct4 (*King and Klose, 2017*; *MacCarthy et al., 2022*). As expected, chromatin accessibility, as assayed by ATAC-seq, scales with Oct4 occupancy, with a sharp dip in accessibility centred at the Oct4 motif likely due to the protection offered by Oct4 binding (*Figure 4—figure supplement 1E*). H3K27ac levels correlate with Oct4 occupancy in regions immediately flanking the Oct4 binding site but are depleted over the binding site itself (*Figure 4—figure supplement 1F*). H3K115ac ChIP profiles also correlate with Oct4 occupancy, but both mononucleosomal and subnucleosomal H3K115ac-marked fragments are enriched over the Oct4 motif itself (*Figure 4—figure supplement 1F*).

As Oct4 binds co-operatively (*Sinha et al., 2023*), we also analysed enhancers with two Oct4 motifs scaling the distance between the two motifs to a window of the same width. The presence of two Oct4 motifs in proximity creates sharp protected footprints in ATAC-seq data, with high chromatin accessibility between the two sites (*Figure 4C*, top panel). The absence of stable nucleosomes between the two motifs is evident in MNase-seq data, with strongly positioned nucleosomes flanking the ATAC-accessible region. H3K27ac is enriched on these flanking nucleosomes but is depleted between the Oct4 motifs (*Figure 4C*, bottom panel). In contrast, enrichment of H3K115ac ChIP signal on both mono- and subnucleosome-sized fragments spans the entire region between the two Oct4 motifs (*Figure 4C*, middle panel).

Active enhancers and promoters are known to harbour fragile nucleosomes carrying the histone variant H3.3 (*Jin and Felsenfeld, 2007*), with the –1 nucleosome reported to be hyperdynamic because of high H3.3 turnover (*Schlesinger et al., 2017*). Consistent with H3K115ac enrichment on the –1 nucleosome (*Figure 3A*), H3K115ac-marked distal elements correlate with sites of high H3.3 turnover (*Deaton et al., 2016*; *Figure 4—figure supplement 1G*). To investigate if H3K115ac then merely correlates with fragile nucleosomes because it is deposited solely on H3.3, we generated a histone H3.3 knock-out mESC line. Immunoblotting shows that global levels of H3K115ac are not affected by H3.3 deletion, indicating that H3K115ac is not specific to H3.3 (*Figure 4D*, *Figure 4—figure supplement 1H*). These data show that H3K115ac and its association with fragile nucleosomes is linked to binding of TF complexes and chromatin accessibility at active enhancer elements and CGI promoters in an H3.3-independent manner.

## H3K115ac is detected at fragile nucleosomes at CTCF binding sites

Though CTCF binding was originally thought to exclude nucleosomes, fragile nucleosomes and subnucleosomal fragments have been detected at CTCF binding sites (*Voong et al., 2016*; *Klein et al., 2023*). MNase-seq data shows two arrays of well-positioned stable nucleosomes on both sides of the CTCF motif, correlated with the level of CTCF occupancy, and consistent with the known ability of CTCF to create arrays of well-positioned nucleosomes (*Fu et al., 2008*; *Owens et al., 2019*; *Figure 5A*). H3K27ac levels are enriched on these +1 and –1 mononucleosomes, low on subnucleosomal-sized fragments, and depleted from the CTCF NDR (*Figure 5—figure supplement 1A*). H3K1115ac levels also correlate with CTCF occupancy. H3K115ac-modified mononucleosomes are enriched at the –1 and +1 positions relative to the CTCF motif, but strikingly H3K115ac-modified subnucleosomal particles are enriched within the CTCF NDR and overlap the CTCF motif (*Figure 5B*).

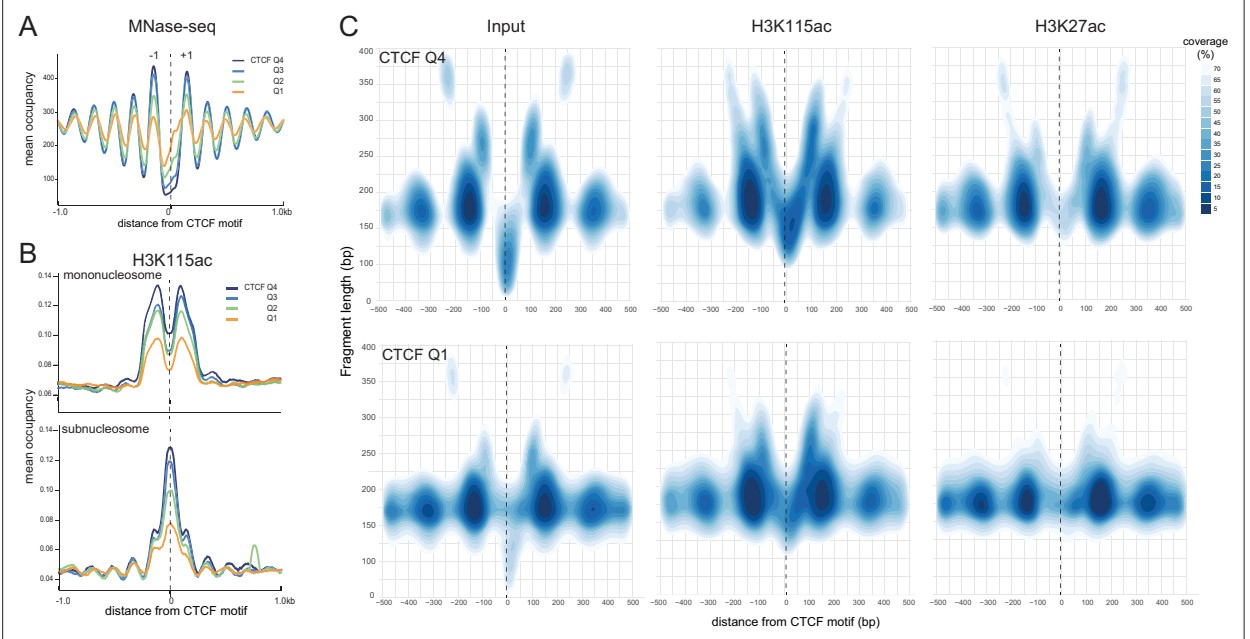

**Figure 5.** H3K115ac marks fragile nucleosomes at sites of high CTCF occupancy. (**A**) Mean occupancy of nucleosomes derived from MNase-seq (from **West et al., 2014**) around CTCF sites across the four quartiles of CTCF ChIP-seq peak strength. All CTCF motifs are oriented from 5' to 3' (left to right). Positions of first flanking upstream (–1) and downstream nucleosome (–1) positions are marked. (**B**) H3K115ac ChIP-seq signal from (top) mononucleosomal- and (bottom) subnucleosomal-sized fragments in mouse embryonic stem cells (mESCs) around CTCF motifs across the four quartiles of CTCF ChIP-seq peak strength as in (**A**). (**C**) Contour plots depicting high-density regions of (left) input, (centre) H3K115ac ChIP, (right) H3K27ac ChIP paired-end sequenced MNase fragments around top (Q4) and bottom (Q1) quartiles of CTCF ChIP-seq peaks in mESCs (data from **Mas et al., 2018**) as a function of fragment length. All CTCF motifs are oriented in the same direction.

The online version of this article includes the following figure supplement(s) for figure 5:

**Figure supplement 1.** H3K27ac and H3K115ac relative to CTCF binding sites and motif orientation.

Using contoured V-plots, we detected small (~50–150 bp) and 250 bp MNase resistant fragments within the CTCF NDR, flanked by strongly positioned nucleosome-sized fragments (compare Q4 to Q1 in *Figure 5C*, left panels). In agreement with average occupancy plots (*Figure 5B*), H5K115ac is detected on the –1 and +1 positioned nucleosomes, and these nucleosomes exhibit a wider range of fragment sizes (*Figure 5C*, middle). This suggests that these nucleosomes are more dynamic than the stable nucleosomes visible in MNase-seq (*Figure 5A*). Within the CTCF NDR, H3K115ac ChIP fragments produce a V-shape pattern, which is more prominent at high CTCF occupancy sites (*Figure 5C*, middle). In the centre of the V, 100–175 bp sized fragments most likely contain H3K115ac-marked subnucleosomes in complex with CTCF. The horns of this V-shaped pattern indicate H3K115ac-marked flanking mononucleosomes interacting with CTCF and/or other factors leading to MNase-protected 225–350 bp fragments. In contrast, H3K27ac is depleted from the NDR, and its enrichment is largely limited to mononucleosomes immediately flanking the CTCF site (*Figure 5C*, right).

V-plots also reveal a subtle asymmetry in the distribution of H3K115ac fragments within the CTCF NDR, with occupancy of subnucleosomal H3K115ac fragments slightly higher towards the 3' end of the CTCF motif. While upstream nucleosomes (MNase-seq) maintain phasing, with decreasing CTCF occupancy, the +1 nucleosome becomes fuzzier and begins to encroach into the CTCF NDR at low CTCF occupancy (*Figure 5—figure supplement 1B*). This directional bias is lost when the orientation of CTCF motifs is randomised. To ascertain that the directional bias is not due to higher CTCF occupancy merely coinciding with TAD boundaries, we used proximity to the nearest TAD boundary (*Bonev et al., 2017*) as the probability with which a given CTCF site participates in TAD boundary formation and divided CTCF motifs into quartiles of this distance (TAD score). We plotted occupancy of stable nucleosomes around the motifs in these quartiles and sorted each quartile by descending order of CTCF occupancy (*Figure 5—figure supplement 1C*). The fuzziness of nucleosomes downstream of CTCF sites is inversely correlated with CTCF occupancy across all TAD scores quartiles.

## Discussion

Lysine acetylation in the globular domain of histones, and particularly at the lateral surface of the nucleosome where histones are in close contact with overlying DNA, is not well studied but is important for nucleosome dynamics, nucleosome stability (*di Cerbo et al., 2014*; *Chatterjee et al., 2015*; *Lawrence et al., 2016*; *Rajagopalan et al., 2017*), and transcription (*Tropberger et al., 2013*). In this study, we explored acetylation of H3K115, a residue located at the nucleosome dyad which weakens histone DNA interactions and destabilises nucleosome in vitro (*Chatterjee et al., 2015*). In *Saccharomyces cerevisiae,* H3K115A and H3K115Q abrogate transcriptional silencing and result in hypersensitivity to DNA damage (*Hyland et al., 2005*). In *Drosophila melanogaster*, both H3K115R and H3K115Q mutations are embryonic lethality (*Graves et al., 2016*), indicating the biological importance of this H3 residue.

We found that in mESCs, and their differentiated derivatives, H3K115ac exhibits a strong preference for active and epigenetically repressed CGI promoters. This is consistent with the known correlation between CGIs and nucleosome depletion independent of transcription (*Tazi and Bird, 1990*; *Fenouil et al., 2012*), their enrichment in the variant histone H2A.Z, and the presence of specific CGI binding proteins that open chromatin (*Yukawa et al., 2014*; *Grand et al., 2021*). In addition to active CGI promoters, H3K115ac is associated with polycomb-repressed promoters in mESCs. Though polycomb complexes can be found at 'bivalent' chromatin domains, characterised by the simultaneous presence of both repressive (H3K27me3) and active (H3K4me3) histone marks at the same promoter regions (*Voigt et al., 2013*), to our knowledge, H3K115ac is the first histone acetylation found within the NDRs associated with polycomb-repressed promoters.

Histone acetylation, particularly in the histone H3 tail, has been used as a marker to identify active enhancers. H3K27ac is most widely used for this purpose (*Moore et al., 2020*; *Stunnenberg et al., 2016*), and various acetyl-lysines in the N-terminal of histone H2B (H2B-NTac) enhance the discovery of active enhancers when used in combination with H3K27ac (*Peng et al., 2020*). However, H3K27ac is dispensable for gene activation in mESCs during the exit from pluripotency and for enhancer chromatin accessibility (*Zhang et al., 2020*; *Sankar et al., 2022*) and it performs poorly as a predictor of functional variants in enhancers (*Biddie et al., 2024*). We previously showed that acetylation of H3K64 and H3K122 in the H3 core marks a subset of active enhancers in mESCs (*Pradeepa et al., 2016*). Whereas H3K115ac marks both active and repressed CGI promoters, we find that H3K115ac is dynamically associated with active enhancers and is enriched within the nucleosome-depleted binding sites of key transcription factors. Its tight association with chromatin accessibility suggests that H3K115ac could enhance detection of active enhancers when used in combination with other PTMs.

A distinct feature of H3K115ac is its association with fragile nucleosome. We show that H3K115ac is enriched in the NDRs associated with promoters, enhancers, and CTCF sites. DNA associated with H3K115ac nucleosomes has higher A/T content than that associated with H3K27ac, consistent with the enhanced MNase sensitivity of nucleosomes associated with A/T-rich sequences (*Chereji et al., 2019*). We confirm the association of H3K115ac with fragile nucleosomes by ChIP-seq of chromatin fractionated on sucrose gradients. In contrast, H3K27ac, H3K64ac, and H3K122ac are depleted from fragile nucleosomes and NDRs. Interestingly, H3K122 and H3K115ac exhibit distinct profiles despite their proximity within the structure of the nucleosome core, suggesting specific and distinct mechanisms of deposition of these marks.

CTCF sites show an H3K115ac pattern distinct from promoters and enhancers as mononucleosomal H3K115ac signal is limited to +1 and −1 positioned nucleosomes flanking CTCF sites while the CTCF NDR is occupied by subnucleosomes marked with H3K115ac most likely in complex with CTCF protein. We note that weakly bound CTCF sites show a weaker nucleosome positioning but only 3' of the CTCF NDR and independent of TAD boundaries. Interestingly, the CTCF N-terminus, which interacts with cohesin, impeding or reversing cohesin-mediated loop extrusion (*Li et al., 2020*; *Nora et al., 2020*), aligns with 3'-end of the CTCF motif (*Yang et al., 2023*). Mechanisms of CTCF binding, orientation, H3K115ac, and nucleosome remodelling may be interlinked.

Though nucleosome fragility has been associated with H3.3 and H2AZ histone variants (*Jin et al., 2009*), we show that H3K115ac is not restricted to H3.3. Our findings support the possibility that acetylation of H3K115 at the dyad axis of the nucleosome enhances the unwrapping or breathing of DNA from the core particle or the release of histone subunits (*Chatterjee et al., 2015*; *Zhou et al., 2019*). The cBAF chromatin remodelling complex has recently been shown to act on fragile nucleosomes at

mESC enhancers to generate hemisomes that facilitate further Oct4 binding (*Nocente et al., 2024*). Given the impact of H3K115 acetylation on nucleosome stability in vitro (*Manohar et al., 2009*; *Simon et al., 2011*) and nucleosome disassembly by chromatin remodelers (*Chatterjee et al., 2015*), it is plausible that H3K115 acetylation facilitates the action of cBAF in further destabilising fragile nucleosomes at enhancers contributing to a dynamic chromatin state. Consistent with our observation of H3K115ac enrichment at repressed CGI promoters in mESCs, polycomb-repressed promoters have also been shown to contain fragile nucleosomes attributed to BAF remodelling (*Brahma and Henikoff, 2024*). To the best of our knowledge, H3K115ac is the first histone modification enriched on a fragile nucleosome at the TSS of CGIs, at polycomb-repressed promoters, at the TF binding sites of active enhancers, and at CTCF sites. H3K115ac is therefore focussed at the heart of regulatory activity in the mammalian genome. In the future, it will be interesting to establish the functional significance of H3K115ac in modulating nucleosome structure, gene regulation, and genome organisation and to ascertain the value of H3K115ac profiling for the functional annotation of genomes.

# Materials and methods

## Key resources table

| Reagent type (species) or resource | Designation | Source or reference | Identifiers | Additional information |
|---|---|---|---|---|
| Cell line (*Mus musculus*) | E14tg2A | *Williamson et al., 2023* | | Mouse embryonic stem cell line |
| Cell line (*Mus musculus*) | 46C Sox1-GFP | *Ying et al., 2003*; *Benabdallah et al., 2019* | | Mouse embryonic stem cell line |
| Cell line (*Mus musculus*) | DPY30 miniAID | *Wang et al., 2023* | | Mouse embryonic stem cell line |
| Other | DMEM/F12 | Gibco | #31330-032 | Cell culture media |
| Other | Neuro basal medium | Gibco | #21103-049 | Cell culture media |
| Other | B27 | Invitrogen | #17504044 | Cell culture media |
| Other | N-2 supplement | Invitrogen | #17502048 | Cell culture media |
| Antibody | Anti-H3K115ac (Rabbit polyclonal IgG) | PTM Bio USA | # PTM-170 | 1:1500 dilution for immunoblot; no dilution for ChIP |
| Antibody | Anti-H3K27ac (Rabbit polyclonal IgG) | Abcam | RRID:AB_2118291 | No dilution for ChIP |
| Antibody | HRP-linked anti-Rabbit IgG (Goat polyclonal) | Cell Signalling Technology | RRID:AB_2099233 | 1:10,000 dilution for immunoblot |
| Antibody | Anti-H3.3 (Rabbit polyclonal) | Millipore | RRID:AB_10845793 | 1:1000 for immunoblot |
| Antibody | Anti-H3 (Rabbit polyclonal) | Abcam | RRID:AB_302613 | 1:5000 for immunoblot |
| Antibody | Anti-H4 (Mouse monoclonal) | Cell Signalling Technology | RRID:AB_1147658 | 1:3000 for immunoblot |
| Antibody | IRDye 800CW Goat anti-Rabbit | LI-COR | t926-32211 | 1:20,000 for immunoblot |
| Antibody | IRDye 680RD Goat anti-mouse | LI-COR | t926-68070 | 1:10,000 for immunoblot |
| Chemical compound, drug | 2-Mercaptoethanol | Gibco | #31350010 | |
| Chemical compound, drug | 4-Thiouridine | Sigma | T4509 | |
| Chemical compound, drug | TRIzol | Invitrogen | 15596026 | |
| Chemical compound, drug | Biotin-HPDP | Pierce | 21341 | |
| Chemical compound, drug | CAIHAKRVTIMK | Thermo Fisher | | Custom H3K115 peptide |
| Chemical compound, drug | CAIHARRVTIMK | Thermo Fisher | | Custom H3K115R peptide |
| Chemical compound, drug | CAIHAK[Ac]RVTIMPK | Thermo Fisher | | Custom H3K115ac peptide |
| Chemical compound, drug | CGGVTIMPK[Ac]DIQLA | Thermo Fisher | | Custom H3K122ac peptide |
| Commercial assay or kit | SuperSignal West Femto Maximum Sensitivity Substrate | Thermo Fisher | # 34096 | HRP detection |
| Commercial assay or kit | 4–20% TGX Gel | Bio-Rad | #4561094 | Pre-cast polyacrylamide gel |
| Commercial assay or kit | SNAP-ChIP K-AcylStat Panel | Epicypher | # 19-3001 | |
| Commercial assay or kit | MNase | New England Biolabs | M0247S | |

*Continued on next page*

*Continued*

| Reagent type (species) or resource | Designation | Source or reference | Identifiers | Additional information |
|---|---|---|---|---|
| Commercial assay or kit | Dynabeads Protein-A | Invitrogen | 10001D | |
| Commercial assay or kit | Tn5 | Illumina | # 20034197 | |
| Commercial assay or kit | AMPure XP beads | Beckman | A63880 | |
| Commercial assay or kit | Qubit assay | Invitrogen | Q32851 | |
| Commercial assay or kit | Turbo DNA-free | Invitrogen | AM1907M | DNase |
| Commercial assay or kit | µMacs Streptavidin beads | Miltenyi | 130-074-101 | |
| Commercial assay or kit | RNeasy MinElute cleanup kit | QIAGEN | 74204 | |
| Commercial assay or kit | RiboMinus eukaryote system v2 | Ambion | A15027 | |
| Commercial assay or kit | NEBNext Ultra II directional RNA library preparation kit | New England Biolabs | E7760 | |
| Commercial assay or kit | NEBNext Ultra II DNA library kit | New England Biolabs | E7645 | |
| Commercial assay or kit | MinElute PCR Purification Kit | QIAGEN | 28004 | |
| Software, algorithm | SRA toolkit (v3.0.5) | https://www.ncbi.nlm.nih.gov/sra | *Kodama et al., 2012* | |
| Software, algorithm | fastp (v0.23.4) | https://github.com/OpenGene/fastp | *Chen et al., 2018*; *Chen, 2026* | |
| Software, algorithm | bowtie2 (v2.5.3) | https://sourceforge.net/projects/bowtie-bio/ | *Langmead and Salzberg, 2012* | |
| Software, algorithm | SAMtools (v1.20) | https://github.com/samtools/samtools | *Danecek et al., 2021*; *Bonfield et al., 2026* | |
| Software, algorithm | Picard (v2.25.1) | https://broadinstitute.github.io/picard/ | http://broadinstitute.github.io/picard/, Broad Institute, |
| Software, algorithm | Bedtools (v2.31.1) | https://bedtools.readthedocs.io/en/latest/ | *Quinlan and Hall, 2010* | |
| Software, algorithm | HOMER (v1.0) | http://homer.ucsd.edu/homer/ | *Heinz et al., 2010* | |
| Software, algorithm | wigTobigWig (v2.9) | https://www.encodeproject.org/software/wigtobigwig/ | *Lee et al., 2022* | |
| Software, algorithm | DESeq2 (v1.50.2) | https://bioconductor.posit.co/packages/3.19/bioc/html/DESeq2.html | *Love et al., 2014* | |
| Software, algorithm | DANPOS3 (v1.0) | https://github.com/sklasfeld/DANPOS3 | *Chen et al., 2013*; *Klasfeld, 2026* | |
| Software, algorithm | deepTools (3.5.1) | https://deeptools.readthedocs.io/en/latest/ | *Ramírez et al., 2016* | |
| Software, algorithm | PeakPredict (v0.1) | https://github.com/efriman/PeakPredict | *Friman, 2024b* | |
| Software, algorithm | matchPWM (v2.78.0) | https://github.com/Bioconductor/Biostrings/blob/devel/R/matchPWM.R | *Pagès et al., 2025*; *Pagès, 2013*; https://doi.org/10.18129/B9.bioc.Biostrings | |
| Software, algorithm | ggdensity (v1.0.1) | https://jamesotto852.github.io/ggdensity/ | *Otto and Kahle, 2023* | |

## Cell culture and neural differentiation

Undifferentiated E14 mESCs were cultured as previously described (*Boyle et al., 2020*). For 46c Sox1-GFP mESCs (*Benabdallah et al., 2016*), differentiation medium (1:1 DMEM/F12: Neuro basal medium, Gibco, #31330-032 and 21103-049, respectively) supplemented with 0.5× B27 (Invitrogen, #17504044), 0.5× N2 (Invitrogen, #17502048), L-glutamine, and 50 mM 2-mercaptoethanol (Gibco, #31350010) was prepared fresh on the day of differentiation. On day 0 (D0), 46c mESCs (*Benabdallah et al., 2016*) were harvested, washed twice with PBS, twice with differentiation media, then seeded at a density of 1×10$^6$ cells on 0.1% gelatin-coated T75 Corning flasks. Cells were cultured for 7 days with daily media changes after day 2. NPC differentiation was monitored visually through expression of the Sox1-GFP reporter.

## Generation of H3.3 KO mESC line

H3.3 KO cells were generated by CRISPR/Cas9 targeting of DPY30-miniAID mESCs (*Wang et al., 2023*), using a combination of two sgRNAs to excise the exon 2 of H3f3a. mESCs were co-transfected

with two eSpCas9 vectors (U6-sgRNA-eSpCas9(1.1)-T2A-mCherry and U6-sgRNA-eSpCas9(1.1)-T2A-GFP) encoding the sgRNA pair using Lipofectamine 3000 according to manufacturer's instructions. GFP+/mCherry+ cells were single cell sorted 48 hr after transfection using a SONY MA900 cell sorter. Resulting clonal cell lines were screened by PCR genotyping for homozygous deletion and confirmed by immunoblotting. Clonal H3f3a knockout cells were used for the next round of H3f3b knockout, using the same strategy to excise the exons 3 and 4 of H3f3b. Sequences of guide RNAs and genotyping primers are in *Supplementary file 4a*.

## Peptide dot blots

Custom peptides (Thermo Fisher Scientific) with the sequence N-CAIHAK115RVTIMPK-C were synthesised as unmodified (H3K115), acetylated K115 (H3K115ac) and lysine to arginine mutation (H3K115R). For H3K122ac, the peptide sequence was N-CGGVTIMPK122DIQLA-C. 2.5, 5, 10, and 20 ng of peptide were spotted on a nitrocellulose membrane and allowed to dry at room temperature (r.t.). The blot was incubated in blocking buffer (5% non-fat dry milk in TBST: 20 mM Tris, 150 mM NaCl, +0.15% Tween-20) for 30 min, followed by primary antibody (1:1500 dilution of anti- H3K115ac, catalogue # PTM170, PTM Bio) in blocking buffer at r.t. before washing 3× with TBST for 5 min each. The blot was then incubated with a 1:10,000 dilution of secondary antibody (HRP-linked anti-Rabbit IgG; Cell Signalling Technology, catalogue # 7074S) in blocking buffer for 45 min at r.t. followed by washing as in previous steps. The blot was developed with HRP substrate (# 34096, Thermo Fisher) and imaged on an ImageQuant LAS-4000 Imager. For peptide sequences, see Key resources table.

## Immunoblotting

Cells were washed with Dulbecco's-PBS, then lysed with 1× Laemmli buffer without 2-mercaptoethanol and Bromophenol blue. The lysates were boiled for 10 min, the protein concentration quantified by BCA assay then adjusted, then 2-mercaptoethanol and Bromophenol blue were added. Lysates were boiled for 5 min prior to loading. Proteins were run on a 4–20% TGX Gel (Bio-Rad) and then transferred to a nitrocellulose membrane (LI-COR). Total protein was visualised by Ponceau staining, then the membrane was blocked with 5% skimmed milk in TBST, rinsed with TBST, and incubated with primary antibodies overnight, and washed with TBST, then incubated with secondary antibodies (LI-COR), washed with TBST, and imaged on LI-COR Odyssey M imager. For antibodies and dilutions, see Key resources table.

## Native ChIP-seq with partial MNase digestion

mESCs and NPCs were harvested with Accutase (Lifetech) and washed twice with cold PBS. 5 million cells were permeabilised by resuspending in 100 µl cold NBA buffer (5.5% sucrose, 85 mM NaCl, 10 mM Tris-HCl pH 7.5, 0.2 mM EDTA, 0.2 mM PMSF, 1 mM DTT, 1X protease inhibitors) and 100 µl of NBB buffer (NBA buffer+0.1% NP-40) and incubated for 10 min on ice. Nuclei were then collected by centrifugation at 1000×*g* for 5 min at 4°C followed by washing with 200 µl NBR buffer (5.5% sucrose, 85 mM NaCl, 10 mM Tris-HCl pH 7.5, 3 mM MgCl$_2$, 1.5 mM CaCl$_2$, 0.2 mM PMSF, 1 mM DTT). Nuclei were resuspended in 100 µl NBR buffer and incubated with 1 µl of RNase-A (10 mg/ml) for 5 min at r.t. followed by treatment with 4 U MNase (Sigma) for 10 min at 20°C while mixing at 500 rpm. MNase was stopped by adding an equal volume of ice-cold 2× STOP buffer (215 mM NaCl, 5.5% sucrose, 10 mM Tris-HCl pH 8, 20 mM EDTA, 2% Triton X-100, 0.2 mM PMSF, 1 mM DTT, 2× protease inhibitors) and placing on ice. The reaction was diluted to 500 µl total volume by adding NBR:STOP buffer and incubated on ice for 14 hr to release digested fragments. Samples were centrifuged at 10,000×*g* for 10 min at 4°C, and solubilised chromatin was recovered as the supernatant. At this point, 10% of the chromatin was saved as Input. Dynabeads Protein-A (Invitrogen) were blocked and pH-calibrated as per the manufacturer's protocol and 5 µl bed volume of beads were loaded with 2 µg of ChIP antibody (H3K27ac; Abcam-AB4729, H3K115ac; PTM BIO-PTM-170). Beads were added to the solubilised chromatin, supplemented with BSA to 0.1 mg/ml, and rotated for 4 hr at 4°C. Beads were collected by a pulse spin and concentrated on a magnetic rack, washed 3× for 5 min each at r.t. with 500 µl ChIP wash buffer (10 mM Tris pH 8, 2 mM EDTA, 150 mM NaCl, 1% NP-40, 1% Na-deoxycholate), followed by a single wash with TE buffer. Bead-bound chromatin was eluted with 100 µl elution buffer (0.1 M NaHCO$_3$, 1% SDS) at 37°C while mixing at 1000 rpm for 15 min. Input chromatin was diluted to 100 µl with elution buffer and the pH

of input and IP samples was adjusted by adding 7 µl of 2 M Tris-Cl pH 6.8 and both were treated with 20 µg Proteinase-K for 2 hr at 55°C while mixing at 1000 rpm. DNA was purified with QIAGEN MinElute columns and sequencing libraries constructed using the NEBNext Ultra II DNA library kit per the manufacturer's instructions. Samples were sequenced for 50 cycles in paired-end mode on an Illumina HiSeq-2000.

## H3K115ac ChIP-qPCR with SNAP-ChIP K-AcylStat panel as spike-in

SNAP-ChIP K-AcylStat Panel from Epicypher (catalogue # 19-3001) provides an equimolar pool of semi-synthetic nucleosome species, each with a single lysine acetylation in combination with a unique DNA barcode. We used this panel as a spike-in as per the manufacturer's instructions and performed native MNase ChIP as described above. After purification of ChIP and Input DNA, qPCR was performed using primers complementary to the barcodes in the K-acyl panel, as well as primer sets targeting the promoter and gene body of murine Klf4 (*Supplementary file 4b*) on a Bio-Rad qPCR machine with 2× SYBR qPCR mix as per the manufacturer's instructions. Recovery of different barcodes was computed as a percent of total input chromatin (*Lin et al., 2012*).

## Sucrose gradient sedimentation of MNase-digested chromatin

Chromatin from unfixed E14 mESCs was prepared as previously described (*Nocente et al., 2024*). Briefly, cells were permeabilised in 15 mM Tris-HCl pH 7.5, 15 mM NaCl, 5 mM $MgCl_2$, 0.1 mM EGTA, 60 mM KCl, 0.3 M sucrose, 0.2% IGEPAL CA-630, and protease inhibitors for 10 min on ice and then centrifuged through an 8 ml sucrose cushion (15 mM Tris-HCl pH 7.5, 15 mM NaCl, 5 mM $MgCl_2$, 0.1 mM EGTA, 60 mM KCl, and 1.2 M sucrose) at 10,000×$g$ for 30 min. Nuclei were resuspended in MNase buffer (20 mM Tris-HCl pH 7.5, 20 mM NaCl, 2 mM $CaCl_2$, 4 mM $MgCl_2$, and 15 mM KCl) and digested for 10 min at 37°C with 1.4 Kunitz units of MNase (New England Biolabs, 200 Kunitz units per µl) per 1×10^6 cells. MNase was stopped by the addition of EDTA to 10 mM and EGTA to 20 mM. After incubation for 1 hr on a rotative agitator at 4°C, chromatin was released by passing the suspension 13 times through a 26-gauge needle, and insoluble material removed by centrifugation. Solubilised chromatin was separated into 500 µl batches derived from approximately 8×10^7 cells. Each batch was loaded onto a 9.9 ml 10–30% sucrose gradient containing 10 mM Tris-HCl pH 7.5, 10 mM EDTA, 20 mM EGTA, and 80 mM NaCl, centrifuged for 16 hr at 197,000×$g$, 4°C, using a Beckman Coulter SW 41 swinging-bucket rotor. 20 fractions of 500 µl were collected from the top to the bottom of the tube.

As previously described (*Nocente et al., 2024*), fractions 5–8 contain subnucleosomal particles of increasing size, fractions 9–10 are enriched in fragile nucleosomes, and fractions 11–13 are composed of stable mononucleosomes. Fractions corresponding to each of these species were pooled, the chromatin quantified by Qbit dsDNA HS assay, and equal amounts of chromatin were taken from each pool and topped up to 1 ml using low salt buffer (10 mM Tris-HCl pH 7.5, 10 mM EDTA, 20 mM EGTA, and 80 mM NaCl) to ensure identical chromatin concentrations in all ChIP reactions. Equal amounts of Spike-In chromatin (from MCF7 cells) were added in the ratio of 1:30, and input samples (10% of total) were saved.

## ATAC-seq sample preparation

ATAC-seq libraries were prepared from 5×10^4 cells as previously described (*Buenrostro et al., 2013*) with some modifications (*Wong et al., 2023*). We used a 15 min incubation on ice in the nuclei preparation step, and the Tn5 reaction was performed in 50 µl of custom transposition buffer (10 mM Tris pH 8, 5 mM $MgCl_2$ and 10% dimethylformamide) with 2.5 µl Tn5 transposase (Illumina, 20034197) at 37°C while mixing at 1000 rpm for 30 min. The reaction was stopped and DNA purified with a MinElute PCR Purification Kit (QIAGEN, 28004). To minimise PCR duplication, we used 20% of tagmented DNA for a test qPCR to determine the minimum number of PCR cycles required to amplify the DNA without entering the plateau phase. The rest of the tagmented DNA was then amplified in identical conditions of dilution and PCR cycles. PCRs were purified and unused adaptors removed with an equal volume of AMPure XP beads (Beckman, A63880) following the manufacturer's protocol and eluted in 20 µl Tris pH 7.8. Libraries were quantified using a Qubit assay (Invitrogen, Q32851) and were selected for the presence of a clear nucleosomal ladder pattern on a Bioanalyzer High Sensitivity assay (Agilent). Samples were sequenced for 50 cycles in paired-end mode on an Illumina HiSeq-2000.

## 4SU sequencing

4SU-seq was performed as described previously (*Boyle et al., 2020*). 4-Thiouridine (4SU; Sigma T4509) was added to cell cultures to a final concentration of 500 μM and incubated for 20 min at 37°C. Cells were collected by trypsinisation, and total RNA purified from 5×10⁶ cells with Trizol (Invitrogen, 15596026) and DNase-treated with a Turbo DNA-free kit (Invitrogen, AM1907M). Thirty micrograms of total RNA were biotinylated using 60 μg of Biotin-HPDP (Pierce, 21341, 1 mg/ml stock in dimethylformamide) in a 300 μl reaction in biotinylation buffer (10 mM Tris-HCl pH 7.4, 1 mM EDTA) for 90 min at r.t. Uncoupled biotin was removed by chloroform extraction (1:1 volume) and labelled RNA precipitated using isopropanol. RNA was resuspended in 100 μl RNAse-free water and incubated with an equal volume of μMacs streptavidin beads (Miltenyi 130-074-101) for 15 min at r.t. Beads were captured on a pre-calibrated μMacs column, followed by three washes with 900 μl of wash buffer pre-heated to 65°C followed by three washes at r.t. RNA was eluted with 100 mM DTT and purified using an RNeasy MinElute cleanup kit (QIAGEN 74204) as per the manufacturer's instructions. Ribosomal RNA was removed using a RiboMinus eukaryote system v2 kit (Ambion, A15027) and stranded libraries constructed using the NEBNext Ultra II directional RNA library preparation kit (NEB, E7760) following the protocol for ribosome-depleted RNA and with an 11 min RNA fragmentation step according to the manufacturer's instructions. Libraries were amplified, purified, and sequenced in paired-end mode as described for ATAC-seq.

## Data analysis

### Quality assessment and pre-processing of sequencing data

Publicly available datasets were downloaded from the NCBI SRA repository with SRA toolkit (fastq-dump). We used FASTQC v0.11.7 for quality control metrics. Filtering of poor-quality reads and trimming of adaptor sequences was performed using fastp (*Chen et al., 2018*) using default settings.

## ChIP-seq analysis

### Single-end ChIP libraries

Filtered and trimmed reads were aligned to the mm10 assembly of the mouse genome using bowtie (*Langmead and Salzberg, 2012*), and uniquely mapped reads were kept. Alignments were converted to BAM format and sorted using SAMtools (*Li et al., 2009*), and PCR duplicates removed using Picard (https://broadinstitute.github.io/picard/). Bedtools (*Quinlan and Hall, 2010*) was used to create BED files followed by conversion into HOMER (*Heinz et al., 2010*) tag directory format (makeTagDirectory) using default parameters. We called peaks using HOMER's findPeaks command with options -F 2 -localSize 50000 -size 150 -minDist 300 -fdr 0.01 –region. UCSC-compatible bigwig files normalised to sequencing depth were generated using makeUCSCfile with parameters: -bigWig -style chipseq –fragLength 150.

### Paired-end ChIP libraries

Filtered and trimmed reads were aligned to the mm10 genome using bowtie2 with options (`--no-discordant --no-mixed --no-unal -X 2000`) followed by conversion to BAM format and removal of PCR duplicates as above. Read pairs were converted into ChIP fragments using bedtools bamtobed –bedpe and fractionated into subnucleosomes (<150 bp) and mononucleosomes (150–225 bp). Fragment coverage was computed at base-pair resolution with bedtools genomeCoverageBed and coverage normalised to sequencing depth and scaled to 10⁷. Coverage data was converted to bigwig format using wigTobigWig (Kent utilities, UCSC). We computed Pearson's correlation between two replicates in 10 kb windows across the genome using the normalised coverage data and replicates were merged based on high correlation (Pearson's coeff.>0.9).

## Differential analysis of ChIP-seq data

To compute significant changes in H3K115ac during ESC to NPC differentiation, we counted reads within 500 bp upstream and downstream of TSS and used these for differential binding in DESeq2 (*Love et al., 2014*). Differential binding is defined as fold change ≥2 and pAdj ≤ 0.01.

## Spike-in normalisation of ChIP-seq libraries

A hybrid genome was created by catenating mouse (mm10) and human (hg38) genome sequences with species-specific chromosome names, and a genome index was prepared using bowtie2-build

command from bowtie2 with default parameters. Reads were pre-processed and aligned to this hybrid genome index as described above for paired-end ChIP libraries. Reads aligning to mouse and human genomes were separated and counted for each sample using SAMtools. A scaling factor was computed for each ChIP and input library as described earlier (*Bressan et al., 2024*):

Scaling factor = $10^6 * (1/\text{number of spike-in reads})$

Read coverage was computed at 10 bp resolution across the mouse genome and multiplied by the scaling factors derived above.

## ATAC-seq analysis

Trimming, filtering, and alignment steps are as described for paired-end ChIP-seq above. The 5′ ends of the reads were offset by +4 bases for reads on the Watson strand, and by −5 bases for the reads on the Crick strand, to reflect the exact location of Tn5 insertion. Peaks were called using macs2 call-peak–nomodel–extsize 150–shift -75 -g 'mm' -p 0.01. Irreproducible discovery rate (≤0.05, *Li et al., 2011*) was used to select reproducible peaks from two biological replicates. For visualisation, a 20 bp fragment was centred on each Tn5 insertion site, and its coverage was calculated at single-base resolution in the whole genome. Scaling and normalisation steps were as for paired-end ChIP-seq above.

## 4SU-seq analysis

Trimming, filtering, and alignment steps are as described for paired-end ChIP-seq. Alignments were converted into HOMER tag directory format with options suitable for stranded libraries – format sam -flip –sspe. Strand-specific genome wide coverage (bigwig format) was computed with parameters: makeUCSCfile –bigwig -fsize 1e20 -strand + (or −) -norm 1e8. Read coverage for promoters was computed using HOMER's annotatePeaks.pl with parameters: -size 'given' -noann -nogene -len 0 -strand both -norm 0. Read counts were then converted into reads per kilobase per million mapped reads. Differential expression analysis was carried out at gene level using the limma package (*Ritchie et al., 2015*).

## MNase-seq data processing and nucleosome calls

Classical MNase-seq (fragments within mononucleosomal range) data for mESCs was downloaded from NCBI SRA (GSE59062, *West et al., 2014*). Trimming, filtering, and alignment steps are as described for paired-end ChIP-seq. High-quality alignments (MAPQ ≥30) from two biological replicates were merged using SAMtools. Nucleosome peaks were called using the DANPOS3 algorithm (*Chen et al., 2015*) with default parameters. Single base resolution wig files from DANPOS3 were converted to bigwig files for visualisation in the UCSC genome browser and plotting coverage using deeptools (*Ramírez et al., 2016*).

## Promoter, CGI, and TSS definitions

Gene annotations were downloaded from the UCSC table browser for genome build: mm10, track: NCBI RefSeq, table: UCSC RefSeq (refGene). For a given TSS (coding and non-coding), 1000 bp upstream and downstream were added and used as a promoter region to filter out promoter proximal peaks in downstream analyses. To create a reference set of unique protein-coding TSSs without duplicated entries, we collapsed the transcripts with identical TSS location. For use in V-plots, a more conservative set of TSSs was defined such that: (i) for genes with multiple TSSs, a single TSS with highest 4SU-seq count in mESCs was retained, (ii) genes with single TSSs were included, (iii) TSSs were filtered out if there was another coding/non-coding TSS within 1 kbp. This ensured that the local and directional observations in our analyses are not affected by transcriptional activity in the vicinity. Association of promoters with CGIs was defined based on a CGI located within 500 bp upstream or downstream of the TSS.

## Random forest model

ATAC-seq accessible non-promoters were overlapped with H3K115ac, H3K122ac, and H3K27ac peaks and classified as H3K115ac-positive or -negative. Peaks from the cistromeDB (*Mei et al., 2017*) and ReMap2022 (*Hammal et al., 2022*) were filtered as described previously (*Friman et al., 2023*). The PeakPredict package (https://github.com/efriman/PeakPredict, copy archived at *Friman, 2024a*) command overlap_peaks was run with settings '`--predict_column H3K115ac --model`

`RandomForestClassifier --balance --shap`', where balancing downsamples the groups to have the same size prior to splitting into test and training sets. The PeakPredict package implements scikit-learn (*Pedregosa et al., 2011*) and SHAP values (*Lundberg and Lee, 2017*).

## CTCF ChIP-seq peaks and motif calls

Analysis of nucleosome positioning around CTCF sites using CTCF ChIP-seq peaks as reference point suffers from low resolution. To mitigate this, we intersected CTCF peaks (*Mas et al., 2018*) in the mESC genome with CTCF motif calls and selected peaks with a single CTCF motif. We used these motif co-ordinates as reference points to plot histone acetylation data. Replicate merged CTCF ChIP-seq peaks for mESCs were downloaded from Remap2022 (*Mas et al., 2018*; *Hammal et al., 2022*; GSE99530). ChIP-seq peak scores were used to split data into quartiles in R. The mouse genome was scanned for consensus CTCF motifs (JASPAR: MA0139.1) using matchPWM in the Biostrings package in Bioconductor. We removed any peaks that overlapped with > 1 CTCF motif and CTCF binding scores divided into quartiles.

## Contour plots

For TSS and CTCF motifs, we plotted signed distance (x-axis) and fragment length (y-axis). We defined an equal number of high-density regions as contours using the package ggdensity in ggplot2 (*Otto and Kahle, 2023*). Contours refer to different levels of fragment density as shown by colour density. Each contour is associated with a probability (%) with which it is bound to data.

## Acknowledgements

We thank Nick Gilbert for discussions. This work has made use of the resources provided by the Edinburgh Compute and Data Facility (ECDF) (http://www.ecdf.ed.ac.uk/) for sequencing data analysis and the Wellcome Trust Clinical Research Facility, Edinburgh (https://clinical-research-facility.ed.ac.uk) for sequencing.

## Additional information

### Funding

| Funder | Grant reference number | Author |
| --- | --- | --- |
| Medical Research Council | MC_UU_00035/7 | Dipta Sengupta |
| Swiss National Science Foundation | P500PB_206805 | Elias T Friman |

The funders had no role in study design, data collection and interpretation, or the decision to submit the work for publication.

### Author contributions

Yatendra Kumar, Conceptualization, Data curation, Software, Formal analysis, Validation, Investigation, Visualization, Methodology, Writing – original draft, Project administration, Writing – review and editing; Dipta Sengupta, Manon Soleil, Hua Wang, Investigation, Methodology; Elias T Friman, Formal analysis, Investigation, Methodology, Writing – review and editing; Robert S Illingworth, Investigation, Writing – review and editing; Zheng Fan, Resources, Investigation, Methodology; Kristian Helin, Supervision, Funding acquisition, Project administration, Writing – review and editing; Matthieu Gérard, Supervision, Funding acquisition, Investigation; Wendy A Bickmore, Conceptualization, Supervision, Funding acquisition, Writing – original draft, Project administration, Writing – review and editing

### Author ORCIDs

Yatendra Kumar ⓘ https://orcid.org/0000-0001-8371-2887
Dipta Sengupta ⓘ https://orcid.org/0000-0002-7836-4774
Elias T Friman ⓘ https://orcid.org/0000-0001-9944-6560
Manon Soleil ⓘ https://orcid.org/0009-0000-4195-8993
Zheng Fan ⓘ https://orcid.org/0000-0003-1934-2521

Hua Wang ⓘ https://orcid.org/0000-0002-1424-0270
Matthieu Gérard ⓘ https://orcid.org/0000-0001-8956-0597
Wendy A Bickmore ⓘ https://orcid.org/0000-0001-6660-7735

Reviewer #1 (Public review): https://doi.org/10.7554/eLife.108802.3.sa1
Reviewer #2 (Public review): https://doi.org/10.7554/eLife.108802.3.sa2
Reviewer #3 (Public review): https://doi.org/10.7554/eLife.108802.3.sa3
Author response https://doi.org/10.7554/eLife.108802.3.sa4

## Additional files

### Supplementary files

Supplementary file 1. Pearson's correlation between biological replicates and sequencing mode (single vs. paired end) for MNase-digested input, H3K27ac and H3K115ac ChIP-seq, and ATAC-seq datasets generated in this study from undifferentiated mouse embryonic stem cells (mESCs) and from mESCs differentiated into neural progenitor cells (NPCs). Data are available in NCBI GEO with the Accession number GSE246191.

Supplementary file 2. Fisher's exact test for data related to *Figure 2*, *Figure 2—figure supplement 1*. (a) Association of gain/loss of H3K115ac with transcriptional changes (up or down) during embryonic stem cell (ESC) to neural progenitor cell (NPC) differentiation for: All promoters, and promoters with or without CpG islands (CGIs). The number of genes is indicated. Data relevant to *Figure 2B*. (b) As in (a) but also for bivalent genes from *Seneviratne et al., 2024*, and including promoters of genes that show no change (nc) in H3K115ac during differentiation. (c) Association of H3K115ac, H3K64ac, H3K122ac, and H3K27ac post-translational modification (PTM) ChIP-seq peaks in mESCs with: All CGI promoters, promoters of bivalent genes from *Seneviratne et al., 2024*, promoters occupied by Ezh2. Data relevant to *Figure 2—figure supplement 1D*.

Supplementary file 3. Wilcox's test for data related to *Figure 3—figure supplement 1*. (a) Statistical data (Wilcox test) for data in *Figure 3—figure supplement 1D* comparing paired-end fragment length between H3K115ac and H3K27ac chromatin immunoprecipitation (ChIP) with the relevant input chromatin (for two biological replicates). (b) Statistical data (Wilcox test) for data in *Figure 3—figure supplement 1E* comparing AT content between ChIP-seq mono- or subnucleosome-sized paired-end fragment lengths for H3K115ac or H3K27ac ChIP data.

Supplementary file 4. Supplementary data for methods. (a) Sequences of (top) guide RNAs to excise the exon 2 of H3f3a and exons 3 and 4 of H3f3b and (below) genotyping primers for H3.3KO DPY30-miniAID mouse embryonic stem cells (mESCs) (*Wang et al., 2023*).

MDAR checklist

### Data availability

The following publicly accessible datasets were used in this study: NCBI GEO GSE66023 (mESC H3K122ac, H3K27ac), GSM1003756 (mESC H3K4me3), GSM1003750 (mESC H3K4me1), GSM1276707 (mESC H3K27me3), GSE59062 (mESC MNase), GSE115774 (mESC 4SU-seq), and GSE78910 (mESC H3.3 turnover). Sequencing data generated in this study are submitted to NCBI GEO with the Accession number GSE246191.

The following dataset was generated:

| Author(s) | Year | Dataset title | Dataset URL | Database and Identifier |
|---|---|---|---|---|
| Kumar Y | 2026 | H3K115 marks fragile nucleosomes at regulatory sites | https://www.ncbi.nlm.nih.gov/geo/query/acc.cgi?acc=GSE246191 | NCBI Gene Expression Omnibus, GSE246191 |

The following previously published datasets were used:

| Author(s) | Year | Dataset title | Dataset URL | Database and Identifier |
|---|---|---|---|---|
| Madapura PM, Grimes G, Bickmore WA | 2016 | Histone H3 globular domain acetylation identifies new class of enhancers | https://www.ncbi.nlm.nih.gov/geo/query/acc.cgi?acc=GSE66023 | NCBI Gene Expression Omnibus, GSE66023 |
| ENCODE DCC | 2012 | Stanford_ChipSeq_ES-E14_H3K4me3_std | https://www.ncbi.nlm.nih.gov/geo/query/acc.cgi?acc=GSM1003756 | NCBI Gene Expression Omnibus, GSM1003756 |
| ENCODE DCC | 2012 | Stanford_ChipSeq_ES-E14_H3K4me1_std | https://www.ncbi.nlm.nih.gov/geo/query/acc.cgi?acc=GSM1003750 | NCBI Gene Expression Omnibus, GSM1003750 |
| Marks H | 2014 | E14_H3K27me3 | https://www.ncbi.nlm.nih.gov/geo/query/acc.cgi?acc=GSM1276707 | NCBI Gene Expression Omnibus, GSM1276707 |
| West JA, Cook A, Alver BH, Stadtfeld M, Deaton AM, Hochedlinger K, Park PJ, Tolstorukov MY, Kingston RE | 2014 | Nucleosome occupancy changes in mammalian cell differentiation and reprogramming (Mnase-Seq) | https://www.ncbi.nlm.nih.gov/geo/query/acc.cgi?acc=GSE59062 | NCBI Gene Expression Omnibus, GSE59062 |
| Benabdallah NS, Williamson I, Illingworth RS, Boyle S, Sengupta D, Therizols P, Bickmore WA | 2019 | Transcriptional profiling by 4SU-seq in mouse ESCs and ESC-derived neural progenitor cells | https://www.ncbi.nlm.nih.gov/geo/query/acc.cgi?acc=GSE115774 | NCBI Gene Expression Omnibus, GSE115774 |
| Deaton AM, Gómez-Rodríguez M, Mieczkowski J, Tolstorukov MY, Kundu S, Sadreyev RI, Jansen LET, Kingston RE | 2016 | Enhancer regions show high histone H3.3 turnover that changes during differentiation | https://www.ncbi.nlm.nih.gov/geo/query/acc.cgi?acc=GSE78910 | NCBI Gene Expression Omnibus, GSE78910 |

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
