## [Editor Report · eLife Assessment]

This **fundamental** study provides the first genome-wide characterization of H3K115 acetylation and identifies a striking and previously unappreciated association of this globular-domain histone modification with fragile nucleosomes at CpG island promoters, active enhancers, and CTCF binding sites. While the work is largely descriptive and correlative in nature the evidence is **compelling**. The authors present multiple, orthogonal genomic and biochemical analyses that consistently support their central conclusions.

---

## [Referee Report · Reviewer #1 (Public review)]

Summary

The authors set out to define the genomic distribution and potential functional associations of acetylation of histone H3 lysine 115 (H3K115ac), a poorly characterized modification located in the globular domain of histone H3. Using native ChIP-seq and complementary genomic approaches in mouse embryonic stem cells and during differentiation to neural progenitor cells, they report that H3K115ac is enriched at CpG island promoters, active enhancers, and CTCF binding sites, where it preferentially localizes to regions containing fragile or subnucleosomal particles. These observations suggest that H3K115ac marks destabilized nucleosomes at key regulatory elements and may serve as an informative indicator of chromatin accessibility and regulatory activity.

Strengths

A major strength of this study is its focus on a histone post-translational modification in the globular domain, an area that has received far less attention than histone tail modifications despite strong evidence from structural and in vitro studies that such marks can directly influence nucleosome stability. The authors employ a wide range of complementary genomic analyses-including paired-end ChIP-seq, fragment size-resolved analyses, contour (V-) plots, and sucrose gradient fractionation-to consistently support the association of H3K115ac with fragile nucleosomes across promoters, enhancers, and architectural elements. The revised manuscript is careful in its interpretation and provides a coherent and internally consistent picture of how H3K115ac differs from other acetylation marks such as H3K27ac and H3K122ac. The datasets generated will be valuable to the chromatin community as a resource for further exploration of nucleosome dynamics at regulatory elements.

Weaknesses

The conclusions are largely correlative. While the authors provide strong evidence for the localization of H3K115ac to fragile nucleosomes, the work does not directly test whether this modification causally contributes to nucleosome destabilization or regulatory function in vivo. Questions regarding the enzymes responsible for depositing or removing H3K115ac and its direct functional consequences therefore remain open.

Overall assessment and impact

Overall, the authors largely achieve their stated aims by providing a detailed and well-supported characterization of H3K115ac distribution in mammalian chromatin and its association with fragile nucleosomes at regulatory elements. While mechanistic insight remains to be established, the study introduces a compelling new perspective on globular-domain histone acetylation and highlights H3K115ac as a potentially useful marker for identifying functionally important regulatory regions. The work is likely to stimulate further mechanistic studies and will be of broad interest to researchers studying chromatin structure, transcriptional regulation, and genome organization.

---

## [Referee Report · Reviewer #2 (Public review)]

Summary:

Kumar et al. aimed to assess the role of the understudied H3K115 acetylation mark, which is located in the nucleosomal core. To this end, the authors performed ChIP-seq experiments of H3K115ac in mouse embryonic stem cells as well as during differentiation into neuronal progenitor cells. Subsequent bioinformatic analyses revealed an association of H3K115ac with fragile nucleosomes at CpG island promoters, as well as with enhancers and CTCF binding sites. This is an interesting study, which provides important novel insights into the potential function of H3K115ac. However, the study is mainly descriptive, and functional experiments are missing.

Strengths:

(1) The authors present the first genome-wide profiling of H3K115ac and link this poorly characterized modification to fragile nucleosomes, CpG island promoters, enhancers, and CTCF binding sites.

(2) The study provides a valuable descriptive resource and raises intriguing hypotheses about the role of H3K115ac in chromatin regulation.

(3) The breadth of the bioinformatic analyses adds to the value of the dataset

Comments on revisions:

The authors sufficiently addressed my concerns.

---

## [Referee Report · Reviewer #3 (Public review)]

Summary:

Kumar et al. examine the H3K115 epigenetic mark located on the lateral surface of the histone core domain and present evidence that it may serve as a marker enriched at transcription start sites (TSSs) of active CpG island promoters and at polycomb-repressed promoters. They also note enrichment of the H3K115ac mark is found on fragile nucleosomes within nucleosome-depleted regions, on active enhancers and CTCF bound sites. They propose that these observations suggest that H3K115ac contributes to nucleosome destabilization and so may servers a marker of functionally important regulatory elements in mammalian genomes.

Strengths:

The authors present novel observations suggesting that acetylation of a histone residue in a core (versus on a histone tail) domain may serve a functional role in promoting transcription in CPG islands and polycomb-repressed promoters. They present a solid amount of confirmatory in silico data using appropriate methodology that supports the idea that H3K115ac mark may function to destabilize nucleosomes and contribute to regulating ESC differentiation. These findings are quite novel.

Weaknesses:

Additional experiments to confirm specificity of the antibodies used have been done, improving confidence in the study.

---

## [Author Response]

The following is the authors’ response to the original reviews.

**Public Reviews:**

**Reviewer #1 (Public reviews):**
(1) The absence of replicate paired-end datasets limits confidence in peak localization.

The reviewer was under the impression that that we did not perform biological replicates of our ChIP-seq experiments. All ChIP-seq (and ATAC-seq) experiments were performed with biological replicates and the Pearson’s correlations (all >0.9) between replicates were provided in Supplementary Table 1. We had indicated this in the text and methods but will try to make this even clearer.

(2) The analyses are primarily correlative, making it difficult to fully assess robustness or to support strong mechanistic conclusions.

Histone modifications are difficult to alter genetically because of the high copy number of histone genes and inhibition of HATs/HDACs in general leads to alterations in other histone modifications. It is an inherent challenge in establishing causality of histone modifications, especially histone acetylation marks.

(3) Some claims (e.g., specificity for CpG islands, "dynamic" regulation during differentiation) are not fully supported by the analyses as presented.

We have modified the text in response to this point. The new text reads: “Non-CGI promoters have lower overall levels of transcription compared to CGI promoters, and for this promoter class H3K115ac enrichment detected by ChIP is only really seen for the highest quartile of transcription (4SU) quartile of expression (Figure 1G). CGI promoters on the other hand, exhibit significant levels of detected H3K115ac even for the lowest quartile of expression. These results suggest a special link between CGI promoters and H3K115ac”.

(4) Overall, the study introduces an intriguing new angle on globular PTMs, but additional rigor and mechanistic evidence are needed to substantiate the conclusions.

We agree that the paper does not provide mechanistic details or solid causality of H3K115ac. We have only emphasized the potential role of H3K115ac in nucleosome fragility based on our in vivo data and previously published in-vitro experiments (Manohar et.al., 2009, Chatterjee et. al., 2015). We do provide the evidence that H3K115ac is enriched on subnucleosomal particles via sucrose gradient sedimentation of MNase-digested chromatin (Figure 3C-D).

**Reviewer #2 (Public review):**
(1) I am not fully convinced about the specificity of the antibody. Although the experiment in Figure S1A shows a specific binding to H3K115ac-modified peptides compared to unmodified peptides, the authors do not show any experiment that shows that the antibody does not bind to unrelated proteins. Thus, a Western of a nuclear extract or the chromatin fraction would be critical to show. Also, peptide competition using the H3K115ac peptide to block the antibody may be good to further support the specificity of the antibody. Also, I don't understand the experiment in Figure S1B. What does it tell us when the H3K115ac histone mark itself is missing? The KLF4 promoter does not appear to be a suitable positive control, given that hundreds of proteins/histone modifications are likely present at this region. It is important to clearly demonstrate that the antibody exclusively recognizes H3K115ac, given that the conclusion of the manuscript strongly depends on the reliability of the obtained ChIP-Seq data.

ChIP-qPCR in S1B includes competition from native chromatin and shows high specificity to its target. We have provided antibody validation in three ways:

- Western blot with dot-blot of synthetic peptides (Figure S1A).

- Western blots with Whole cell extracts (Figure 4D).

- ChIP-qPCR on native chromatin spiked with a cocktail of synthetic mono-nucleosomes, each carrying a single acetylation and a specific barcode (SNAP-ChIP K-AcylStat Panel).

We could not include H3K115ac marked nucleosomes as they are not available in the panel. Figure S1B shows that the H3K115ac antibody exhibits negligible binding to known K-acyl marks, comparable to an unmodified nucleosome. Because of the absence of a H3K115ac modified barcoded nucleosome, we used the KLF4 promoter from mESCs as a positive control, in agreement with ChIP-seq signal shown in the genome browser profile (Figure 1E), the KLF4 promoter shows a significantly higher signal than the gene body.

(2) The association of H3K115ac with fragile nucleosomes is based on MNase-sensitivity and fragment length, which are indirect methods and can have technical bias. Experiments that support that the H3K115ac modified nucleosomes are indeed more fragile are missing.

We have performed ChIP-seq on MNase digested mESC chromatin fractionated on sucrose gradients and this shows that H3K115ac is enriched in fractions containing sub-nucleosomal and fragile nucleosomes but depleted in fractions containing stable nucleosomes (Figure 3D).

(3) The comparison of H3K115ac with H3K122ac and H3K64ac relies on publicly available datasets. Since the authors argue that these marks are distinct, data generated under identical experimental conditions would be more convincing. At a minimum, the limitations of using external datasets should be discussed.

H3K64ac and H3K122ac datasets were generated by us in a previous publication (Pradeepa et. al., 2016) using same native MNase ChIP protocol as used here. The ChIP-seq datasets for H3K122ac and H3K27ac are processed in an identical manner, with the same computational pipelines, to the H3K115ac data sets generated in this paper.

(4) The enrichment of H3K115ac at enhancers and CTCF binding sites is notable but remains descriptive. It would be interesting to clarify whether H3K115ac actively influences transcription factor/CTCF binding or is a downstream correlate.

We agree with the reviewer’s comment, but we have not claimed causality.

(5) No information is provided about how H3K115ac may be deposited/removed. Without this information, it is difficult to place this modification into established chromatin regulatory pathways.

Due to broad target specificity, redundancies and crosstalk among different classes of HATs and HDACs, it is not tractable to answer this question in the current manuscript.

**Reviewer #3 (Public reviews):**

Reviewer 3 is mistaken in thinking our ChIP experiments are performed under cross-linked conditions. As clearly stated in the main text and methods, all our ChIP-seq for histone modifications is done on native MNase-digested chromatin – with no cross-linking. This includes the spike-in experiment shown in Fig S1B to test H3K115ac antibody specificity against the bar-coded SNAP-ChIP K-AcylStat Panel from Epicypher. We could not include H3K115ac bar-coded nucleosomes in that experiment since they are not available in the panel.

**Recommendations for the authors:**

**Reviewer #1 (Recommendations for the authors):**
(1) I have two primary concerns that resound through the entire paper:(a) Overall, the manuscript is making strong claims based on entirely correlative datasets. No quantitative analyses are performed to demonstrate co-occupancy/localization. Please see more detailed descriptions below.

Our responses to specific points are provided against each comment below.

(b) Lack of paired-end replicates for H3K115ac ChIP-seq. While the reviewer token for the deposited data was not made accessible to me, looking at Supplementary Table 1, it appears there are two H3K115ac ChIP-seq datasets. One is paired-end and is single-read. So are peaks called with only one replicate of PE? Or are inaccurate peaks called with SR datasets? Either way, this is not a rigorous way to evaluate H3K115ac localization.

We are sorry that this reviewer was not able to access the data – the token for the GEO accession was provided for reviewers at the journal’s request. All ChIP-seq (and ATAC-seq) experiments (paired and single-end) were performed with two biological replicates and the Pearson’s correlations (all >0.9) between replicates were provided in Supplementary Table 1. This was indicated in both the main text and in the methods. In the revised manuscript we have tried to make this even clearer and have put the relevant Pearsons coefficient (r) into the text at the appropriate places. For the reviewer’s information, here is the complete list of data samples in the GEO Accession:

**Author response image 1. sa4fig1:** 

While I agree that H3K115ac occupancy is high at +CGIs, the authors downplay that H3K122ac and H3K27ac is also more highly enriched at these locations (page 7, last sentence of first paragraph). I imagine this is all due to the more highly transcribed nature of these genes. Sub-stratifying the K27ac and K122ac by transcription (as in Figure 1G) would help to demonstrate a unique nature of H3K115ac. But even better would be to do an analysis that plots H3K115ac enrichment vs transcription for every individual gene rather than aggregate analyses that are biased by single locations. For example, make an XY scatterplot of RNAPII occupancy or 4SU-seq signal vs H3K115ac level, where each point represents a single gene. Because the interpretation that it is CGI-based and not transcription is confounded with the fact that -CGI are more lowly transcribed. So, looking at Figure 1G, even the -CGI occupancy of H3K115ac is correlated with transcription, but it is just more lowly transcribed.

We thank the reviewer for these suggestions but point out that Figure 1G shows H3K115ac signal for CGI+ and CGI– TSS that are matched for expressions levels (quartiles of 4SU-seq). Fig 1F shows that H3k115ac is much more of a discriminator between CGI+ and – than H3K27ac or H3K122ac.

(2) H3K115ac, H3K27ac, and H3K122ac are all more enriched (in aggregate) at +CGI locations (Fig 1F); so do these locations just have more positioned nucleosomes? More H3.3? So that these PTMs are just more enriched due to the opportunity?

Positioned nucleosomes are generally found downstream of the TSS of active CpG island promoters, so what the reviewer suggests may well account for the relative enrichment of H327ac and H3K122ac at CGI+ vs CGI- promoters in Fig.1F. But H3K115ac localisation is distinct, with the peak at the nucleosome-depleted region not the +1 nucleosome. This is also confirmed by the contour plots in Fig 3. Our observation is also not explained by an enrichment of H3.3 at CGI promoters, since we show that H3K115ac is not specific to H3.3 (Fig 4D).

(3) The authors note in paragraph 2 of page 7 that "H3K115ac does not scale linearly with gene expression..." but the authors never show a quantification of this; stratification in four clusters is not able to make a linear correlation. Furthermore, in the second line of page 7, the authors state that the levels do generally correlate with transcription. To claim it is a specific CGI link and not transcription is tricky, but I encourage the authors to consider more quantifiable ways, rather than correlations, to demonstrate this point, if it is observed.

We thank the reviewer for this comment, and taking it into consideration, we have decided to re-phrase this paragraph. The new text reads: “Non-CGI promoters have lower overall levels of transcription compared to CGI promoters, and for this promoter class H3K115ac enrichment detected by ChIP is only really seen for the highest quartile of transcription (4SU) quartile of expression (Figure 1G). CGI promoters on the other hand, exhibit significant levels of detected H3K115ac even for the lowest quartile of expression. These results suggest a special link between CGI promoters and H3K115ac”.

(4) The authors claim on page 7 that "on average, transcription increased from TSS that also gained H3K115ac but to a modest extent, compared with the more substantial loss of H3K115ac from downregulated TSS". However, both upregulated and downregulated are significant; the difference in magnitude could simply be due to more highly or more lowly transcribed locations, meaning that fold change could be more robustly detected. I caution the authors to substantiate claims like this rather than stating a correlation.

We thank the reviewer for this comment which relates to the data in Fig 2A. It is Fig. 2B shows that the association of H3K115ac loss with downregulation is statistically stronger than H3K115ac gain with upregulation, but only for CGI promoters. With regard to the text on the original pg 7 that is referred to, we have now reworded this to read “Average levels of transcription increased from TSS that also gained H3K115ac, and there was loss of H3K115ac from downregulated TSS (Figure 2A).”

(5) For Figure 2C, the authors argue that H3K115ac correlate with bivalent locations. So this is all qualitative and aggregate localization; please quantitatively demonstrate this claim.

Figure S2D provides statistics for this (observed/expected and Fishers exact test).

(6) The authors claim in Figure 2 that H3115ac is dynamic during differentiation (title of Figure 2). However, there are locations that gain and lose, or maintain H3K115ac. In fact, the most discussed locations are H3K115ac with no change (2C); which means it is NOT dynamic during differentiation. So what is the message for the role during differentiation? From Supplemental Table 1, it appears there is a single ChIP experiment for H3K115ac in NPC, and it is a single read. So this is also a difficult claim with one replicate. Related to this, in S2A, the authors show K115ac where there is no change in transcription; so what is the role of H3K115ac at TSSs relevant to differentiation - it is at both locations changed and unchanged in transcription, but H3K115ac levels itself do not change at these subsets. So, how is this dynamic? This is very confusing, and clearer analyses and descriptions are necessary to deconvolute these data.

We apologise for the misleading title for Figure 2. This has now been amended to “Changes in H3K115ac during differentiation”. The message of this figure is that whilst changes in H3K115ac at TSS are small (panels A-C), at enhancers the changes are much more dramatic (panel D). The reviewer is incorrect about the number of replicates for NPCs – there are two biological replicates (see response to point 1b).

(7) The authors go on to examine H3K115ac enrichment on fragile nucleosomes through sucrose gradient sedimentation. A control for H3K27ac or H3K122ac would be nice for comparison.

We do not have the material available to perform these experiments

(8) When discussing Figures 3 and SF3, the authors mention performing a different MNase for a second ChIP. Showing the MNase distribution for both the more highly digested and the lowly digested would be nice. (a) Related to the above, the authors show input in SF3E to argue that the difference in H3K115ac vs H3K27ac is not due to the library, but they do not show the MNase digestion patterns, which is more important for this argument.

Input libraries (first two graphs of FigS3E) are the MNase-digested chromatin. Comparison of nucleotide frequencies from millions of reads is more robust method than the fragment length patterns.

(9) The authors move on to examine H3K115ac at enhancers. Just out of curiosity, given what was found at promoters, is H3K115ac enriched at +CGI enhancers? And what is the correlation with enhancer transcription?

This is an interesting point, but the number of enhancers associated with CGI is not very high and so we did not focus on this. We have not analysed a correlation with eRNAs in this paper.

(10) The authors state on page 14 that the most frequent changes in H3K115ac during differentiation are at these enhancers. So do these changes connect with differentiation-specific genes, and/or genes that have altered transcription during differentiation? Just trying to understand the functional role.

Given the challenges of connecting enhancers with target genes, we have not addressed this question quantitatively. However, we draw the reviewer’s attention to the Genome Browser shots in Figures 2D and S2C, which show clear gain of H3K115ac (and ATAC-seq peaks) at intra and intergenic regions close to genes whose transcription is activated during the differentiation to NPCs.

(11) Related, at the end of page 14, the authors state that the changes in H3K115ac correlate with changes in ATAC-seq; I imagine this dynamic is not unique for H3K115ac and this is observed for other PTMs (H3K27ac), so assessing and clarifying this, to again get to the specific interest of H3K115ac, would be ideal.

We have not claimed that chromatin accessibility is unique to H3K115ac. It is the location of H3K115ac which is found inside the ATAC-seq peak region while H3K27ac is found only upstream/downstream of the ATAC peak that is so striking. This is apparent in Fig 4C.

(12) The authors examine levels of H3K115ac in H3.3 KO cell lines via western blot (Figure 4D), but no replicates and/or quantification are shown.

We now provide a biological replicate for the Western Blot (new FigS4H) together with an image of the whole gel for the data in Fig 4D

(13) In Figure S4 and at the end of page 17, the authors are arguing that there is a link to pioneer TF complexes, based on Oct4 binding. First, while Oct4 has pioneering activity, not all Oct4 sites (or motifs) are pioneering; this has been established. So if you want to use Oct4, substratifying by pioneer vs no pioneer is necessary. Second, demonstrating this is unique to pioneer and not to non-pioneer TFs would be an important control.

In response to the reviewer’s comment, we have removed the term “pioneer” from the manuscript.

(14) Minor point: Figure 4 A and B, there are some formatting issues with the scale bars.

We thank the reviewer for pointing this out, and the errors have been corrected in the revised figure.

(15) Minor point is that it should be clear when single replicates of data are used and when PE/SR sequences are combined or which one is used in each analysis, as this was hard to discern when reading the paper and figure legends.

We have clearly stated in the text that, after Figure2, we repeated all experiments in paired-end mode. All processing steps are defined separately for single end and paired end datasets in the method section. Details of biological replicates are provided in Sup. Table 1. These concerns are also addressed in our response to Reviewer’s public comment-1.

(16) Minor point: it is surprising that different MNase and different units were used in the ChIP vs sucrose sedimentation. Could the authors clarify why?

Chromatin prep for sucrose gradients were done on a much larger scale than for ChIP-seq and required different setups to obtain the right level of MNase digestion.

(17) The authors note that fragile nucleosomes contain H2A.Z and H3.3, but they never perform an analysis of available data to demonstrate a correlation (or better a quantifiable correlation) between H3K115ac occupancy and these marks at the locations they identify H3K115ac.

Since have shown (Fig. 4) that depletion of H3.3 does not affect overall levels of H3K115ac, we do not think there is value in further quantitative correlative analyses of H3K115ac and variant histones.

(18) Minor point: What is the overlap in peaks for H3K115ac, H3K122ac, and H3K27ac (Figure 1C)?

Nearly all H3K115ac peaks overlap with H3K122ac and/or H3K27ac. Its most distinct properties are its association with CGI promoters, fragile nucleosomes and its unique localisation within the NDRs, three points that the manuscript is focussed on.

**Reviewer #3 (Recommendations for the authors):**
(1) The western blot results in Figure 4D probing for H3, H3.3, and H3K115ac use Ponceau S staining, presumably of an area of the membrane where histones might be expected to migrate, as a measure of loading. However, the Ponceau S bands appear uniformly weaker in the H3.3KO lanes, yet despite this, blotting with H3.3 antibody detects a band in H3.3 knockout ESCs, suggesting that the antibody does not have a high degree of specificity. Again, a blocking experiment with appropriate peptides would instill more confidence in the specificity of these reagents, and/or the authors could provide independent validation of the knockout model to differentiate between a partial knockout or antibody cross-reactivity (e.g., by Sanger sequencing).

In a revised Fig. S4H we now show the whole gel corresponding to this blot but including co-staining with an antibody for H4 to provide a better loading control. We also provide a biological replicate of this Western blot in the lower panel of Fig. S4H.

(2) The manuscript would benefit from in vitro follow-up and validation, but if the authors intend to keep the manuscript primarily in silico, I suggest dedicating a few lines in each section to explain the plots, their axes, and their purpose, as well as to assist with interpretation, rather than directly discussing the results. This would make the manuscript more accessible and understandable for a broader audience in the field of epigenetics.

In the revised version, we have tried to improve the text to make the data more accessible to a broad audience.